# Matriptase activation of Gq drives epithelial disruption and inflammation via RSK and DUOX

Jiajia Ma[1], Claire A Scott[2,3], Ying Na Ho[1], Harsha Mahabaleshwar[1], Katherine S Marsay[2,4], Changqing Zhang[1], Christopher KJ Teow[1], Ser Sue Ng[2], Weibin Zhang[2], Vinay Tergaonkar[2], Lynda J Partridge[4], Sudipto Roy[2,5,6], Enrique Amaya[3], Tom J Carney[1,2]*

[1]Lee Kong Chian School of Medicine, Experimental Medicine Building, Yunnan Garden Campus, 59 Nanyang Drive, Nanyang Technological University, Singapore, Singapore; [2]Institute of Molecular and Cell Biology (IMCB), A*STAR (Agency for Science, Technology and Research), Singapore, Singapore; [3]Division of Cell Matrix Biology and Regenerative Medicine, School of Biological Sciences, Faculty of Biology, Medicine and Health, University of Manchester, Manchester, United Kingdom; [4]Department of Molecular Biology and Biotechnology, University of Sheffield, Sheffield, United Kingdom; [5]Department of Biological Sciences, National University of Singapore, Singapore, Singapore; [6]Department of Pediatrics, Yong Loo Ling School of Medicine, National University of Singapore, Singapore, Singapore

*For correspondence:
tcarney@ntu.edu.sg

Competing interests: The authors declare that no competing interests exist.

**Abstract** Epithelial tissues are primed to respond to insults by activating epithelial cell motility and rapid inflammation. Such responses are also elicited upon overexpression of the membrane-bound protease, Matriptase, or mutation of its inhibitor, Hai1. Unrestricted Matriptase activity also predisposes to carcinoma. How Matriptase leads to these cellular outcomes is unknown. We demonstrate that zebrafish *hai1a* mutants show increased $H_2O_2$, NfκB signalling, and IP$_3$R -mediated calcium flashes, and that these promote inflammation, but do not generate epithelial cell motility. In contrast, inhibition of the Gq subunit in *hai1a* mutants rescues both the inflammation and epithelial phenotypes, with the latter recapitulated by the DAG analogue, PMA. We demonstrate that *hai1a* has elevated MAPK pathway activity, inhibition of which rescues the epidermal defects. Finally, we identify RSK kinases as MAPK targets disrupting adherens junctions in *hai1a* mutants. Our work maps novel signalling cascades mediating the potent effects of Matriptase on epithelia, with implications for tissue damage response and carcinoma progression.

## Introduction

The transmembrane serine protease, Matriptase, encoded by the *ST14* gene, has potent oncogenic properties and is consistently dysregulated in human carcinomas. Overexpression of Matriptase in the mouse epidermis leads to epidermal papillomas, ulcerative and invasive carcinomas, and inflammation (*List et al., 2005*; *Martin and List, 2019*). These effects of Matriptase are mitigated by a cognate serine protease inhibitor, HAI-1. Clinically, an increase in the Matriptase:HAI-1 ratio has been found in a range of tumours and is predictive of poor outcome (*Martin and List, 2019*). Loss of mouse Hai1 leads to epidermal and intestinal barrier defects, epithelial inflammation, and failure of placental labyrinth formation, which are all due to unrestricted Matriptase activity (*Kawaguchi et al., 2011*; *Nagaike et al., 2008*; *Szabo et al., 2007*). The response of epithelia to

**eLife digest** Cancer occurs when normal processes in the cell become corrupted or unregulated. Many proteins can contribute, including one enzyme called Matriptase that cuts other proteins at specific sites. Matriptase activity is tightly controlled by a protein called Hai1. In mice and zebrafish, when Hai1 cannot adequately control Matriptase activity, invasive cancers with severe inflammation develop. However, it is unclear how unregulated Matriptase leads to both inflammation and cancer invasion.

One outcome of Matriptase activity is removal of proteins called Cadherins from the cell surface. These proteins have a role in cell adhesion: they act like glue to stick cells together. Without them, cells can dissociate from a tissue and move away, a critical step in cancer cells invading other organs. However, it is unknown exactly how Matriptase triggers the removal of Cadherins from the cell surface to promote invasion.

Previous work has shown that Matriptase switches on a receptor called Proteinase-activated receptor 2, or Par2 for short, which is known to activate many enzymes, including one called phospholipase C. When activated, this enzyme releases two signals into the cell: a sugar called inositol triphosphate, IP3; and a lipid or fat called diacylglycerol, DAG. It is possible that these two signals have a role to play in how Matriptase removes Cadherins from the cell surface.

To find out, Ma et al. mapped the effects of Matriptase in zebrafish lacking the Hai1 protein. This revealed that Matriptase increases IP3 and DAG levels, which initiate both inflammation and invasion. IP3 promotes inflammation by switching on pro-inflammatory signals inside the cell such as the chemical hydrogen peroxide. At the same time, DAG promotes cell invasion by activating a well-known cancer signalling pathway called MAPK. This pathway activates a protein called RSK. Ma et al. show that this protein is required to remove Cadherins from the surface of cells, thus connecting Matriptase's activation of phospholipase C with its role in disrupting cell adhesion.

An increase in the ratio of Matriptase to HAI-1 (the human equivalent of Hai1) is present in many cancers. For this reason, the signal cascades described by Ma et al. may be of interest in developing treatments for these cancers. Understanding how these signals work together could lead to more direct targeted anti-cancer approaches in the future.

unregulated Matriptase activity appears conserved across vertebrates. Mutation of the zebrafish orthologue, Hai1a, also results in epidermal defects, including loss of membrane E-cadherin, aberrant mesenchymal behaviour of keratinocytes, which form cell aggregations over the body and loss of fin fold integrity. The epidermis also displays sterile inflammation and is invaded by highly active neutrophils. Genetic ablation of the myeloid lineage demonstrated that the keratinocyte phenotypes are not a consequence of the inflammation (*Carney et al., 2007*). The strong *hai1a*[fr26] allele is embryonic lethal, dying within the first week, whilst the more mild allele, *hai1a*[hi2217], is semi-viable, with epithelial defects resolved through sphingosine-1-phosphate-mediated entosis and cell extrusion (*Armistead et al., 2020*). All *hai1a* mutant phenotypes can be ameliorated by reduction of Matriptase levels (*Carney et al., 2007*; *Mathias et al., 2007*).

Due to the clinical implications of its dysregulation, the signalling pathways activated pathologically by Matriptase are of interest. The G-protein-coupled receptor, proteinase-activated receptor-2 (Par2), is essential for the oncogenic and inflammatory effects of uninhibited Matriptase in zebrafish and mouse (*Sales et al., 2015*; *Schepis et al., 2018*). Par2 is directly activated by Matriptase proteolysis and signals through a number of heterotrimeric Gα protein subunits. Early studies in keratinocytes linked Par2 activation with intracellular $Ca^{++}$ mobilisation via phospholipase C, thus implicating Gq subunit as the canonical target (*Schechter et al., 1998*). Alternate Gα subunits, including Gi, Gs, and G12/13, are now known to also be activated by Par2 (*Zhao et al., 2014*). Par2 displays biased agonism, and the logic of the pathway utilised depends on cell context and the activating protease. In vitro experiments using HEK293 cells implicated both Par2 and Gi in Matriptase-mediated Nfκb pathway activation (*Sales et al., 2015*). Whilst this explains the inflammatory phenotype of uninhibited Matriptase, it does not address whether Par2 promotes carcinoma phenotypes directly in keratinocytes in vivo. In zebrafish, as the keratinocyte defects are not dependent on inflammation, but are dependent on Par2, it is likely that there is a direct effect of Par2 on promoting keratinocyte motility.

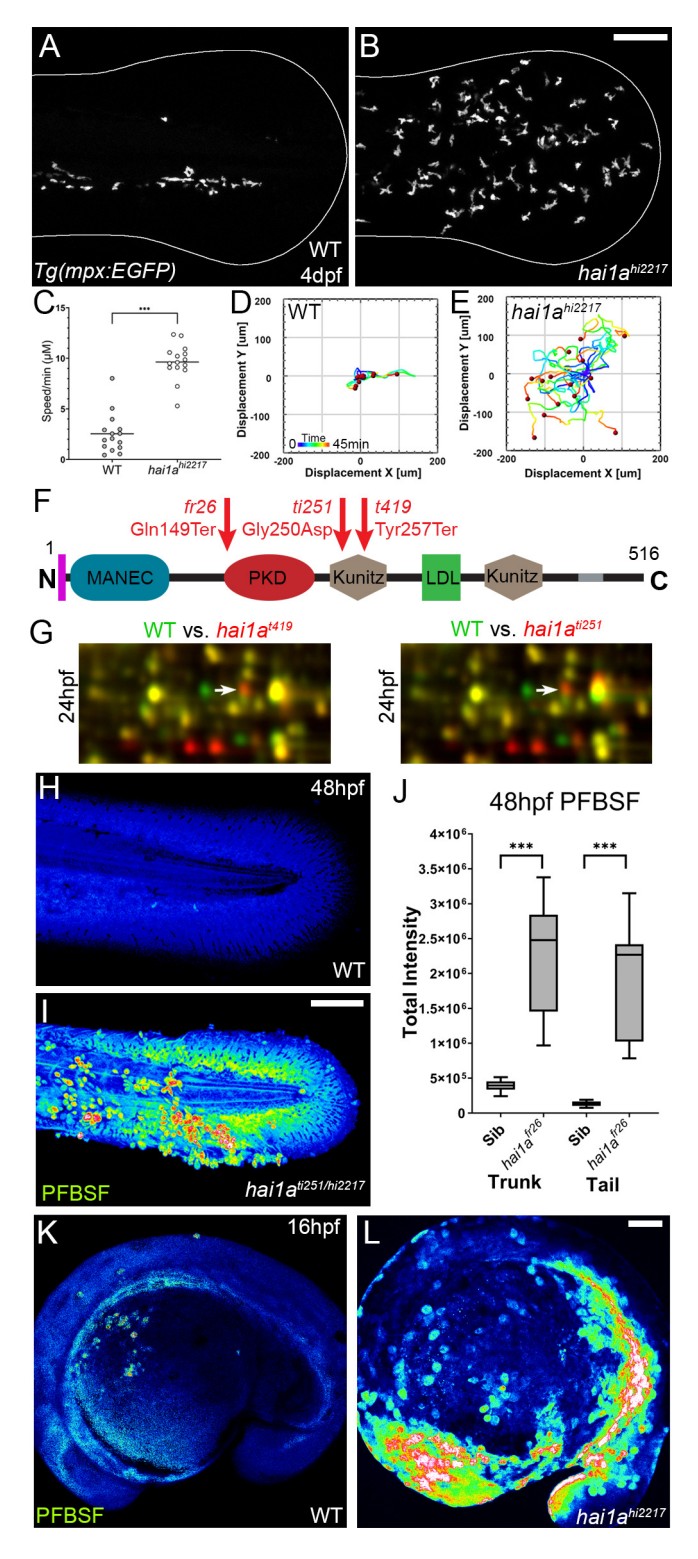

**Figure 1.** The epidermis of *hai1a* mutants displays elevated hydrogen peroxide. (**A, B**) Projected confocal images showing neutrophils populate the tail of *hai1a*<sup>hi2217</sup> mutants (**B**) but just the vasculature of WT (**A**) at 4dpf labelled by the *Tg(mpx:EGFP)*<sup>i114</sup> line. Fin extremity outlined in white. (**C**) Neutrophils move significantly faster in *hai1a*<sup>hi2217</sup> than WT. n = 15; t-test; ***p<0.001. (**D, E**) Tracks of neutrophil migration taken from *Video 1* in WT (**D**) and *hai1a*<sup>hi2217</sup> (**E**). (**F**) Schematic of the Hai1a protein with protein domains given, signal peptide as purple line and

*Figure 1 continued on next page*

*Figure 1 continued*

transmembrane domain as grey line. Location and nature of the *fr26* and two *dandruff* alleles, *ti251* and *t419* given. (G) Selected region of 2D gel of protein extracted from 24hpf embryos for *hai1a^t419^* (left) or *hai1a^ti251^* (right) in red, superimposed over WT protein samples (green in both). The shift in pI of peroxiredoxin4 in both alleles is indicated with an arrow. (H, I) Projected lateral confocal views of pentafluorobenzenesulfonyl fluorescein (PFBSF) staining of WT (H) and *hai1a^ti251/hi2217^* (I) tail fins at 48hpf. (J) Box and whiskers plot of PFBSF fluorescent staining intensity of WT and *hai1a^fr26^* mutants at 48hpf in trunk and tail. n = 9; t-test \*\*\*p<0.001. (K, L) Projected lateral confocal views of PFBSF staining of WT (K) and *hai1a^hi2217^* (L) at 16hpf. Scale bars: (B, I, L) = 100 µm.

The online version of this article includes the following source data and figure supplement(s) for figure 1:

**Source data 1.** 2D proteomics protein ID report list of the spot identities with significant ratio changes.
**Source data 2.** 2D proteomics protein ratio changes for each of the protein spots identified as significantly changed.
**Source data 3.** 2D proteomics top 50 proteins significantly changed in *hai1a* mutants.
**Figure supplement 1.** The epidermis of *hai1a* mutants displays elevated hydrogen peroxide.

Par2 can also transactivate EGFR through an unknown mechanism, and inhibition of EGFR alleviates certain basal keratinocyte phenotypes of zebrafish *hai1a* mutants (*Schepis et al., 2018*). Thus, the identity, contribution, and interactions of the pathways downstream of Matriptase and Par2 remain unclear. Here through use of the zebrafish *hai1a* mutant, we comprehensively map the essential pathways downstream of zebrafish Matriptase and Par2, leading to inflammation and epithelial disruption.

## Results

### Increased hydrogen peroxide and calcium flashes contribute to inflammation in *hai1a* mutants

Neutrophils in *hai1a* embryos invade the epidermis, are highly motile, but move randomly (*Carney et al., 2007*; *Mathias et al., 2007*; *Figure 1A–E*, *Video 1*). To establish the nature of their stimulus, we tested if neutrophils in *hai1a* altered their behaviour in the presence of a large fin wound. In wild-type larvae, neutrophils were recruited from a great distance and tracked to the wound with high directionality. However, neutrophils in the *hai1a* mutant appeared largely apathetic to the wound and remained migrating randomly. There was a mild increase in neutrophil speed in *hai1a* larvae following wounding, indicating that they retain capacity to respond to additive stimuli (*Figure 1—figure supplement 1A–D*, *Video 2*). Co-labelling of basal keratinocyte nuclei (using TP63 immunostaining), neutrophils (*Tg(fli1:EGFP)^y1^* transgenic), and TUNEL labelling of apoptotic cells highlighted that whilst the epidermis of *hai1a* mutants, unlike WT, had regions of apoptosis at 24hpf (arrowhead, *Figure 1—figure supplement 1E, F*), neutrophils were not associated, but rather found at nascent TUNEL-negative aggregates of basal keratinocytes (arrow). We conclude that epi-dermal cell death does not directly contribute to inflammation and that the effector stimulating neutrophils in *hai1a* mutants is as, or more, potent as that of wounds.

To identify the neutrophil activator in *hai1a*, we employed an unbiased approach using 2D gel proteomics to compare the wild-type proteome with that of strong *hai1a* alleles. The *dandruff (ddf)* mutant has many phenotypic similarities to the strong *hai1a^fr26^* allele (*van Eeden et al., 1996*). Crosses between *ddf^ti251^* or *ddf^t419^* and *hai1a^hi2217^* failed to complement, and sequencing of *hai1a* cDNA from both *ddf* alleles identified a nonsense mutation in the *ddf^t419^* allele (c.771T>G; p.Tyr257Ter) and a missense mutation of a highly conserved amino acid in the *ddf^ti251^* allele (c.749G>A; p.Gly250Asp) (*Figure 1F*,

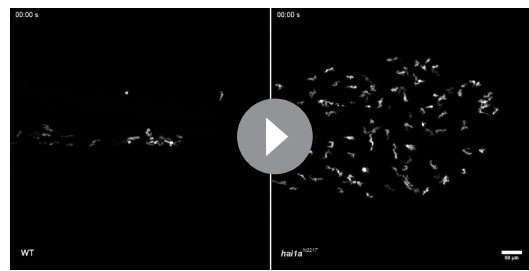

**Video 1.** Neutrophils in WT and *hai1a^hi2217^* 4dpf larva. Projected confocal timelapses of eGFP-positive neutrophils in the tail region of 4dpf *Tg(mpx:eGFP)^i114^* (left) and *hai1a^hi2217^*; *Tg(mpx:eGFP)^i114^* (right) larvae taken every 45 s for 45 min. Scale bar: 50 µm.
https://elifesciences.org/articles/66596#video1

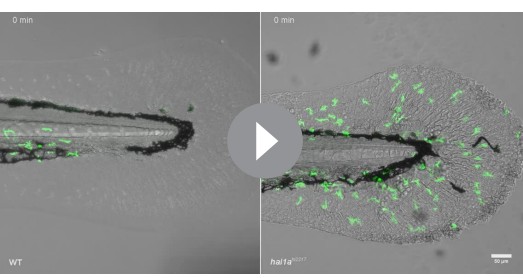

**Video 2.** Neutrophils in WT and *hai1a*^*hi2217*^ 4dpf larva before and after fin wound. Projected confocal timelapses of eGFP-positive neutrophils in the tail region of 4dpf *Tg(mpx:eGFP)*^*i114*^ (left) and *hai1a*^*hi2217*^; *Tg(mpx:eGFP)*^*i114*^ (right) larvae taken every 50 s for 250 min with the tail fin cut at 50 min. GFP is overlaid on DIC (Differential Interference Contrast) channel. Scale bar: 50 μm.

https://elifesciences.org/articles/66596#video2

*Figure 1—figure supplement 1G–I*). We used both alleles for comparative 2D protein gel analysis at 24hpf and 48hpf. Rather than finding proteins with altered molecular weight, Peroxiredoxin4 (Prdx4) was identified as having a higher pI in both *hai1a*^*t419*^ and *hai1a*^*ti251*^ mutants at 24hpf and 48hpf, indicative of a change in oxidation state (*Figure 1G*, *Figure 1—figure supplement 1J, K*). Peroxiredoxins are hydrogen peroxide scavengers, and its altered oxidation state suggested that *hai1a* has higher $H_2O_2$ levels, a known activator of inflammation in larval zebrafish (*Niethammer et al., 2009*). Pentafluorobenzenesulfonyl fluorescein (PFBSF) staining *Maeda et al., 2004* demonstrated significantly higher levels of $H_2O_2$ in the trunk and tails of *hai1a* mutants at 24hpf and 48hpf (*Figure 1H–J*, *Figure 1—figure supplement 1L, M*). This increase in $H_2O_2$ in *hai1a* was observed as early as 16hpf, and thus preceded presentation of *hai1a* phenotypes (*Figure 1K, L*).

To demonstrate that, as with other phenotypes, the $H_2O_2$ increase in *hai1a* was due to unrestrained activity of Matriptase1a, we used a *matriptase1a* mutant allele, *st14a*^*sq10*^, which prematurely terminates the protein at 156 amino acids (*Figure 2A*, *Figure 2—figure supplement 1A–C*; *Lin et al., 2019*). Zygotic *st14a* mutants showed no overt phenotype; however, maternal zygotic mutants lacked ear otoliths (*Figure 2B, C*). As expected, when crossed into the *hai1a* background, embryos lacking otoliths (*st14a*^*sq10*^; *hai1a*^*hi2217*^ double mutants) never displayed the *hai1a* epidermal and neutrophil phenotypes (*Figure 2D–F*; *Table 1*). Double mutants also had significantly reduced $H_2O_2$ levels (*Figure 2F*, *Figure 2—figure supplement 1D*). To determine if reduced $H_2O_2$ could account for the rescue of *hai1a* phenotypes by *st14a* mutation, we used genetic and pharmacological inhibition of the main enzyme responsible for generating $H_2O_2$ in zebrafish, Duox. A morpholino directed against *duox* successfully reduced $H_2O_2$ levels (*Figure 2*, *Figure 2—figure supplement 1D*) and neutrophil inflammation in *hai1a* mutants but did not rescue the epithelial defects (*Figure 2F, G*). Treatment with a known Duox inhibitor, diphenyleneiodonium (DPI), also resulted in amelioration of neutrophil inflammation, but not epithelial aggregates, in *hai1a* mutants (*Figure 2G*, *Figure 2—figure supplement 1E*). We conclude that Matriptase1 activity leads to excess $H_2O_2$ in *hai1a* mutants, which partially accounts for the neutrophil inflammation, but not epidermal defects.

Duox is regulated by calcium through its EF-Hand domains, and calcium flashes are known to generate $H_2O_2$ in epidermal wounds in *Drosophila* (*Razzell et al., 2013*). We injected *hai1a*^*fr26*^ with RNA encoding the calcium reporter *GCaMP6s*. Timelapse imaging at 24hpf indicated that *hai1a* mutants had significantly more calcium flashes in both the trunk and tail (*Figure 3A, B, E*, *Figure 3—figure supplement 1A, B*, *Video 3*). Increased intracellular calcium dynamics was observable as early as 16hpf, concomitant with increased $H_2O_2$, but prior to onset of *hai1a* phenotypes (*Figure 3G, H*, *Video 4*). Release of calcium from intracellular stores is regulated by $IP_3$ receptors located on the endoplasmic reticulum. The frequency and number of calcium flashes in the trunk and tail of *hai1a* mutants are reduced by treatment with the $IP_3R$ inhibitor, 2-APB compared to control (*Figure 3C, D, F*, *Figure 3—figure supplement 1C, D*, *Video 5*). Reducing calcium flashes in *hai1a* mutant embryos with 2-APB also significantly reduced $H_2O_2$ levels (*Figure 3I, J*, *Figure 3—figure supplement 1E*) and partially reduced inflammation; however, the epidermal defects were not noticeably rescued (imaged by DIC (Differential Interference Contrast) or labelled with the TP63 antibody) (*Figure 3I–K*). We observed similar reduction in neutrophil inflammation, but not rescue of epidermal defects, in *hai1a* mutants treated with thapsigargin, which inhibits the replenishment of ER calcium stores by SERCA (*Figure 3K*, *Figure 3—figure supplement 1F, G*). This suggests, in *hai1a* mutants, that $IP_3R$-dependent calcium flashes activate Duox, flooding the epidermis with $H_2O_2$ and leading to inflammation.

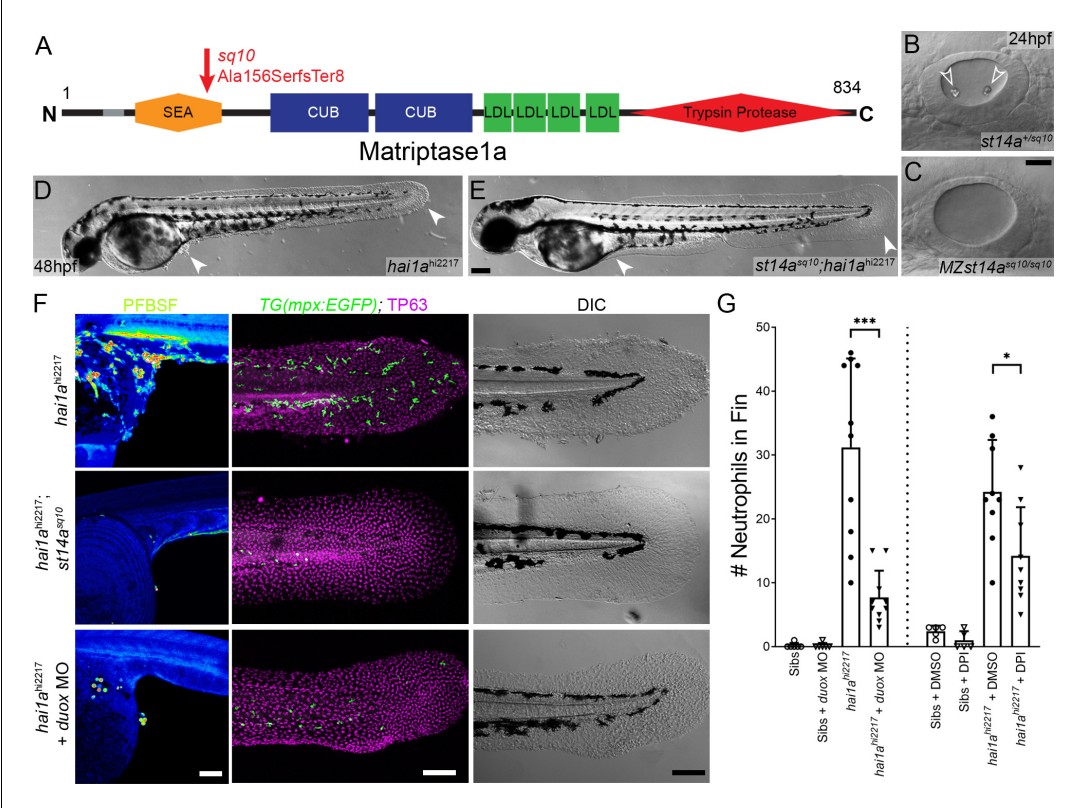

**Figure 2.** Loss of Matriptase1a or Duox1 reduces H$_2$O$_2$ and neutrophils in *hai1a* mutants. (**A**) Schematic of the Matriptase1a protein with domains given and transmembrane domain as grey line. Location and nature of the *sq10* allele given by red arrow. (**B, C**) Lateral DIC (Differential Interference Contrast) images of *st14a$^{+/sq10}$* (**B**) and MZ *st14a$^{sq10}$* (**C**) otic vesicles at 24hpf showing absence of otoliths (arrowheads in **B**) in the maternal zygotic *st14a* mutants. (**D, E**) Lateral DIC images of *hai1a$^{hi2217}$* single mutant (**D**) and *st14a$^{sq10}$; hai1a$^{h12217}$* double mutant (**E**) at 48hpf highlighting rescue of epidermal aggregates and fin morphology (arrowheads) in the double mutants. (**F**) Projected confocal images of pentafluorobenzenesulfonyl fluorescein (PFBSF) staining at 24hpf (left column), TP63 (magenta), and eGFP (green) antibody staining at 48hpf (middle column) with DIC imaging (right column) for *hai1a$^{hi2217}$* single mutants (top row), *st14a$^{sq10}$; hai1a$^{hi2217}$* double mutants (middle row), and *hai1a$^{hi2217}$* mutants injected with 0.4 mM, *duox* MO + 0.2 mM *tp53* morpholino (bottom row). Individuals for middle and right columns were hemizygous for the *Tg(mpx:eGFP)$^{i114}$* transgene. (**G**) Counts of eGFP-positive neutrophils on the fins of *hai1a$^{hi2217}$; Tg(mpx:eGFP)$^{i114}$* or *Tg(mpx:eGFP)$^{i114}$*, and either uninjected, injected with morpholino against *duox* (left side of graph), treated with 0.5% DMSO (Dimethyl sulfoxide) or 40 µM diphenyleneiodonium (DPI) (right side of graph). n = 10; t-test; ***p<0.001; *p<0.05. Scale bars: (**C**) = 20 µm; (**E, F**) = 100 µm.

The online version of this article includes the following figure supplement(s) for figure 2:

**Figure supplement 1.** Generation of *st14a* mutant and Duox inhibition reduces hydrogen peroxide and neutrophils in *hai1a* mutants.

## Hydrogen peroxide elevates NfkB signalling in *hai1a* mutants

Increased Matriptase, Par2 activity, or hydrogen peroxide levels are known to activate NfkB signalling (*Kanke et al., 2001*; *Sales et al., 2015*; *Schreck et al., 1991*). We crossed the *hai1a$^{fr26}$* allele to the NfkB sensor transgenic line *Tg(6xHsa.NFKB:EGFP)$^{nc1}$*. In WT embryos, NfkB signalling was

**Table 1.** Prevalence of otolith and epithelial phenotypes in *hai1a* and *st14a* double mutants: p<0.0001 (Chi-squared test).

*hai1a$^{+/hi2217}$; st14a$^{+/sq10}$* ♂ × *hai1a$^{hi2217/hi2217}$; st14a$^{sq10/sq10}$* ♀

| Observed (*expected*) | WT epidermis | *hai1a* epidermis | Total |
|---|---|---|---|
| Wild-type otoliths | 72 (*65*) | 60 (*65*) | 132 (*130*) |
| No otoliths | 128 (*65*) | 0 (*65*) | 128 (*130*) |
| Total | 200 (*130*) | 60 (*130*) | 260 |

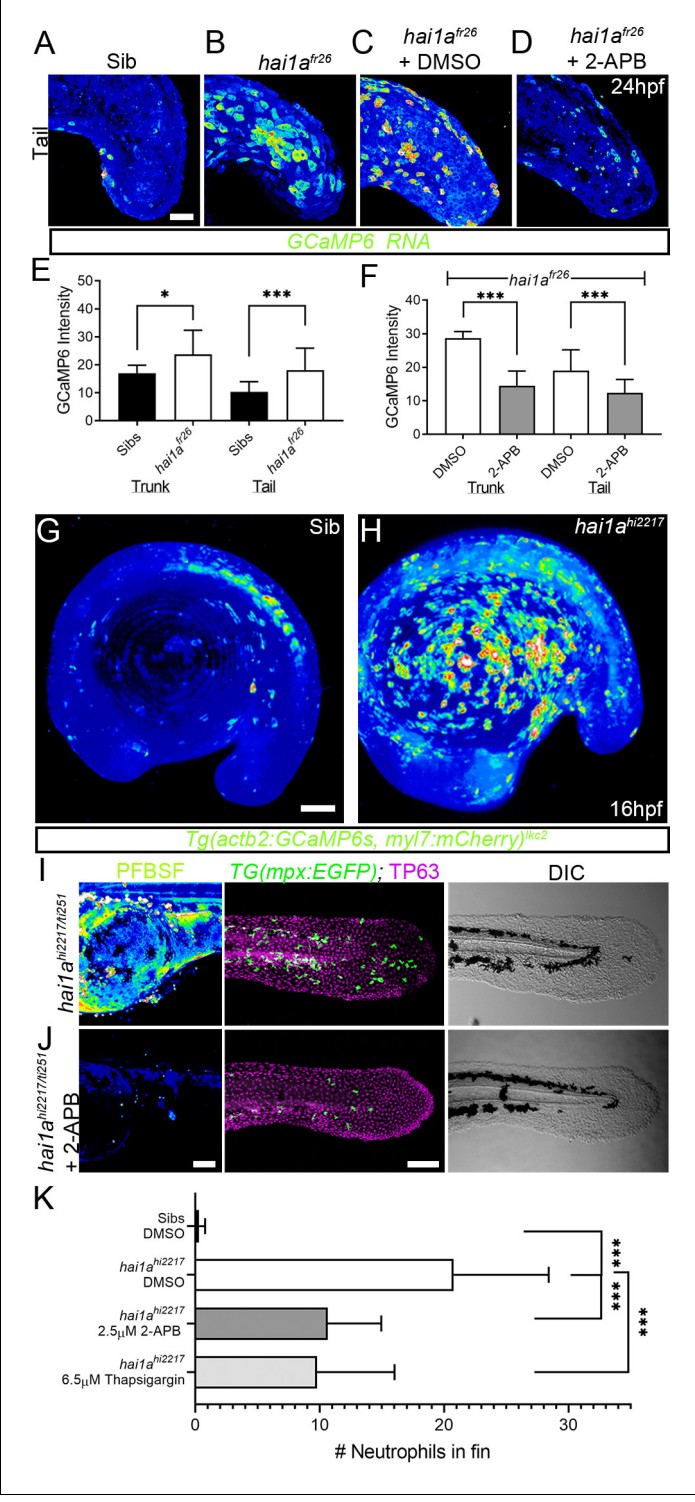

**Figure 3.** Calcium dynamics in *hai1a* mutants regulate $H_2O_2$ and inflammation. (A–D) Projected confocal images of eGFP in the tail of WT (A) or *hai1a^fr26*; (B–D) injected with *GCaMP6s* RNA, imaged at 24hpf, indicating calcium dynamics. Embryos are either untreated (A, B), treated with DMSO (C), or with 2.5 µM 2-APB (D). Images are temporal projections of timelapse movies taken at maximum speed intervals (2 min) and projected by time. (E, F) Graphs comparing eGFP intensities from *GCaMP6s* RNA timelapses in trunk and tail between 24hpf WT and *hai1a^fr26* (E) and between *hai1a^fr26* treated with DMSO and 2.5 µM 2-APB (F). n = 10; t-test; *p<0.05, ***p<0.001. (G, H) Projected light-sheet images of *Tg(actb2:GCaMP6s, myl7:mCherry)^lkc2* embryos indicating calcium dynamics at 16hpf of sibling (G) or *hai1a^hi2217* (H). Images are temporal projections of timelapse movies taken at 45 s

*Figure 3 continued on next page*

*Figure 3 continued*

intervals and projected by time. (I, J) Pentafluorobenzenesulfonyl fluorescein (PFBSF) staining at 24hpf (left column), TP63 (magenta), and eGFP (green) antibody staining at 48hpf (middle column) with DIC imaging (right column) for *hai1a*<sup>hi2217/ti251</sup> mutants (J), or *hai1a*<sup>hi2217/ti247</sup> mutants treated with 2.5 µM 2-APB (I). Individuals for middle and right columns were hemizygous for the *Tg(mpx:eGFP)*<sup>i114</sup> transgene. (K) Counts of eGFP-positive neutrophils in the fins at 48hpf of *Tg(mpx:eGFP)*<sup>i114</sup>, or *hai1a*<sup>hi2217</sup>; *Tg(mpx:eGFP)*<sup>i114</sup> treated with 0.5% DMSO, 2.5 µM 2-APB, or 6.5 µM thapsigargin. n = 20; t-test; ***p<0.001. Scale bars (A–D) = 50 µm; (G, I, J) = 100 µm. The online version of this article includes the following figure supplement(s) for figure 3:

**Figure supplement 1.** Thapsigargin and 2-APB reduce neutrophil inflammation but not epidermal defects in *hai1a* mutants.

---

mostly restricted to neuromasts at 48hpf, whilst in *hai1a* mutants we observed an increase in fluorescence at 24hpf and a strong increase at 48hpf. Fluorescence at both timepoints was noted in epidermal aggregates and fin folds, locations of strong inflammation (*Figure 4A, B*, *Figure 4—figure supplement 1A, B*). This increase in signalling in 48hpf *hai1a* mutant embryos was confirmed by qRT-PCR of the NfkB target gene, *nfkbiaa* (*Figure 4C*). Unlike calcium and $H_2O_2$, NfkB signalling is not present at early stages prior to phenotype (*Figure 4—figure supplement 1C, D*). To determine the extent that NfkB signalling accounts for the *hai1a* phenotypes, we generated a mutant in the *ikbkg* (=*ikkg* or *nemo*) gene, which encodes a scaffold protein required for activating the NfkB pathway (*Rothwarf et al., 1998*) (*ikbkg*<sup>sq304</sup> Gly80ValfsTer11; *Figure 4—figure supplement 1E*). Crossing this mutant to *hai1a*<sup>hi2217</sup> on the *Tg(mpx:eGFP)*<sup>i114</sup> background realised a very strong rescue of neutrophil inflammation at 48hpf, but no improvement of *hai1a* epidermal defects (*Figure 4D–I*). To demonstrate that this increase in NfkB signalling was dependent on $H_2O_2$, we injected *hai1a*<sup>hi2217</sup>; *Tg(6xHsa.NFKB:EGFP)*<sup>nc1</sup> embryos with *duox* MO. We noted a strong reduction in NfkB pathway activation compared to uninjected *hai1a*<sup>hi2217</sup> mutant controls (*Figure 4J, K*). Conversely, genetic ablation of NfkB signalling, through use of the *ikbkg* mutant, did not reduce $H_2O_2$ levels in *hai1a* mutants (*Figure 4—figure supplement 1F, G*). Similarly, we tested if reduction of calcium flashes could also reduce NfkB signalling in *hai1a* mutants using 2-APB but noticed only a slight reduction (*Figure 4—figure supplement 1H, I*). We propose that upon loss of Hai1a, $IP_3$R-mediated release of calcium activates Duox to increase $H_2O_2$. This acts upstream of NfkB pathway activation, which occurs at later stages, and is necessary for the inflammation phenotype.

## Both inflammation and epidermal aggregates of *hai1a* mutants are resolved by Gq inhibition

$IP_3$ is generated from cleavage of $PIP_2$ by Phospholipase C. The sensitivity of the *hai1a* mutants to 2-APB implies that $IP_3$ levels are increased and therefore there may be an increase in Phospholipase C activity. Numerous attempts to inhibit PLC in *hai1a* mutants failed, and we were unable to find a dosage window that rescued without gross embryo deformity. Hence, we tested rescue of *hai1a* mutants with YM-254890, an inhibitor of the heterotrimeric G protein alpha subunit, Gq, which directly activates PLC isoforms. We found that not only did this significantly reduce neutrophil inflammation (*Figure 5D, F*), but surprisingly, it also significantly rescued the epidermal defects in *hai1a* mutants, with a significant reduction in TP63-positive epidermal aggregates in the trunk and improved tail fin fold integrity at 48hpf (*Figure 5A–E*).

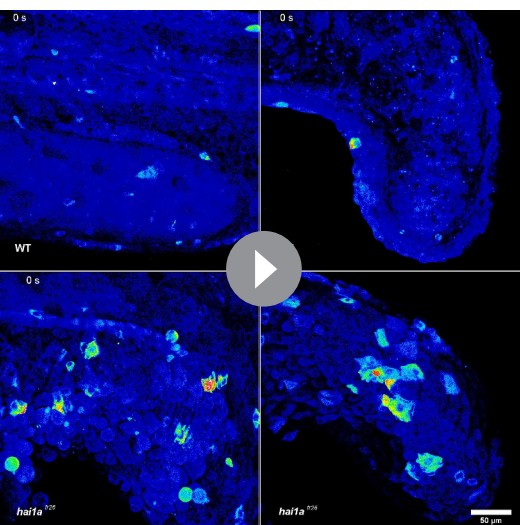

**Video 3.** Calcium dynamics in WT and *hai1a*<sup>fr26</sup> embryos at 24hpf. Projected confocal timelapses of eGFP in the trunks (left side) and tails (right side) of a 24hpf WT (top row) and *hai1a*<sup>fr26</sup> (bottom row) embryos injected with *GCaMP6s* RNA, indicating calcium dynamics. Scale bar: 50 µm.
https://elifesciences.org/articles/66596#video3

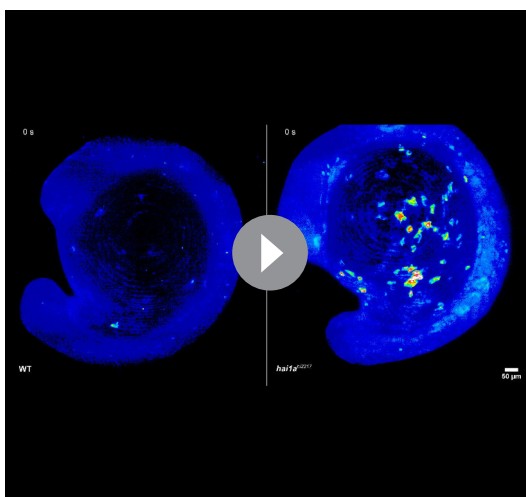

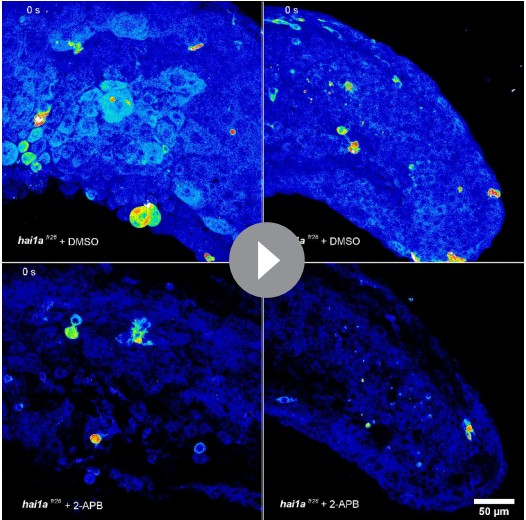

**Video 4.** Calcium dynamics in WT and *hai1a^hi2217^* embryos at 16hpf. Projected light-sheet timelapses of eGFP in WT (left side) and *hai1a^hi2217^* (right side) embryos at 16hpf. Both embryos carried the *Tg(actb2: GCaMP6s, myl7:mCherry)^lkc2^* transgene reporting calcium dynamics, which were higher in the *hai1a* mutant, particularly over the yolk. Images were taken every 45 s for 19 min. Scale bar: 50 µm.
https://elifesciences.org/articles/66596#video4

**Video 5.** Calcium dynamics in DMSO and 2-APB-treated *hai1a^fr26^* embryos at 24hpf. Projected confocal timelapses of eGFP signal in the trunks (left side) and tails (right side) of 24hpf *hai1a^fr26^* embryos injected with *GCaMP6s* RNA and treated with 0.03% DMSO (top row) and 2.5 µM 2-APB (bottom row), indicating reduced calcium dynamics following 2-APB treatment. Scale bar: 50 µm.
https://elifesciences.org/articles/66596#video5

## PMA treatment phenocopies the *hai1a* mutant

As IP$_3$R inhibition only blocks inflammation in *hai1a* mutants, but an inhibitor of a PLC activator (Gq) additionally reduces the epidermal defects, we considered that diacyl glycerol (DAG) might contribute to the epidermal defects as the second product of PIP$_2$ cleavage (along with IP$_3$). Indeed, treating WT embryos from 15hpf to 24hpf with 125 ng/ml phorbol 12-myristate 13-acetate (PMA), a DAG analogue, resulted in embryos with striking similarities to strong *hai1a* mutants, including a thin or absent yolk sac extension, lack of head straightening, lack of lifting the head off the yolk, and multiple epidermal aggregates on the skin (*Figure 6A–C*). These aggregates were due, at least partially, to displacement of basal keratinocytes as shown by TP63 staining where the basal keratinocyte nuclei lost their uniform distribution (*Figure 6D, E*). Treatment from 24hpf to 48hpf with 125 ng/ml PMA led to a fin defect similar to the dysmorphic *hai1a* mutant fin (*Figure 6F, G*). It has been shown that the basal keratinocytes in *hai1a* lose their epithelial nature and adopt a partially migratory phenotype (*Carney et al., 2007*; *Video 6*). We treated *Tg(krtt1c19e:lyn-tdtomato)^sq16^* larvae (*Lee et al., 2014*) with 37.5 ng/ml PMA for 12 hr and imaged the basal epidermis at 3dpf by light-sheet timelapse. Whilst the DMSO-treated transgenic larvae had very stable keratinocyte membranes and shape, PMA treatment led to a less stable cell membrane topology and dynamic cell shape, similar to *hai1a* mutants (*Figure 6H*, *Videos 7* and *8*). Kymographs taken from *Video 7* highlighted both the more labile and weaker cell membrane staining following PMA treatment (*Figure 6I*). The potency of PMA was dependant on region and reduced with age.

Most PMA-treated *Tg(mpx.eGFP)^i114^* larvae at 48hpf also had more neutrophils in the epidermis than untreated controls, which were highly migratory (*Figure 6F–G, J–K'*, *Video 8*). We determined H$_2$O$_2$ levels in PMA-treated embryos using PFBSF staining and found that it was significantly increased in both trunk and tail at 24hpf (*Figure 6L–O, R*). In contrast, when we treated *GCaMP6s* RNA-injected embryos with PMA, we failed to see an increase in calcium flashes, as seen in *hai1a* (*Figure 6P, Q, S*). To see if the heightened H$_2$O$_2$ and inflammation was also correlated with increased NfkB signalling, we treated *Tg(6xHsa.NFKB:EGFP)^nc1^* embryos with 125 ng/ml PMA. There

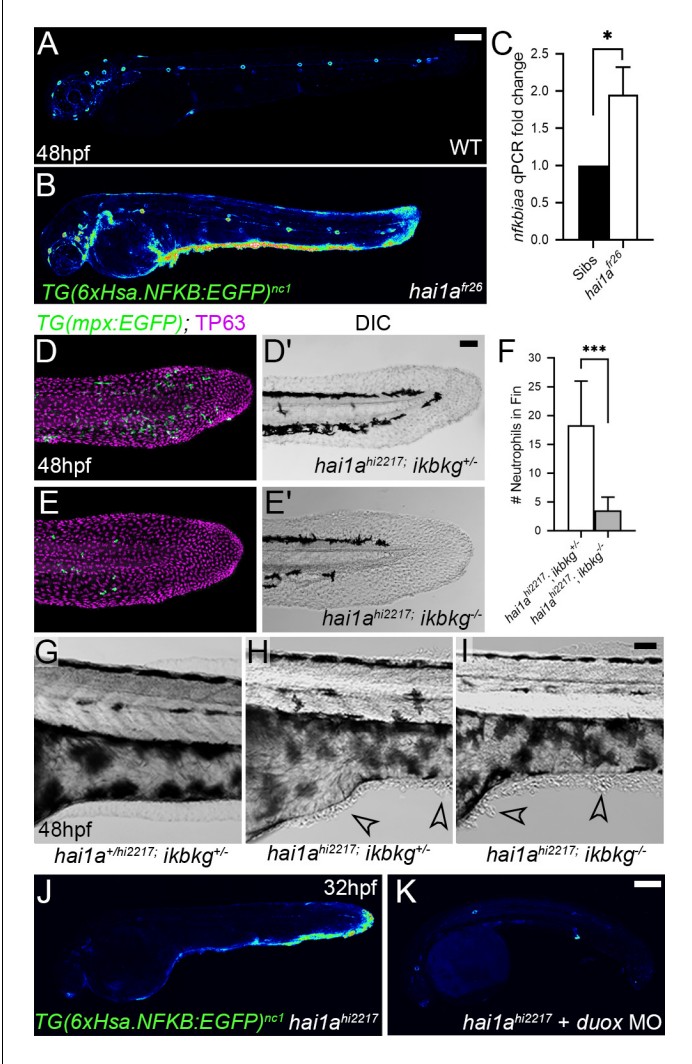

**Figure 4.** NfkB signalling is elevated in *hai1a* mutants and is necessary for neutrophil inflammation. (**A, B**) Lateral confocal projections of *Tg(6xHsa.NFKB:EGFP)^nc1* embryos reporting NfkB signalling levels at 48hpf for WT (**A**) and *hai1a^fr26* (**B**). (**C**) qPCR of cDNA levels of NfkB target gene *nfkbiaa* in *hai1a^fr26* vs. sibs at 48hpf. n = 3, 200 embryos pooled in each, t-test *p<0.05. (**D–E′**) Projected confocal images of the tail fins of 48hpf *Tg(mpx:eGFP)^i114*; *hai1a^hi2217* embryos, immunostained for TP63 (magenta) and eGFP (green) (**D, E**) with corresponding DIC image (**D′, E′**). Embryos were either mutant for *ikbkg* (*ikbkg^sq304*, **E–E′**) or heterozygous (*ikbkg^+/sq304*; **D–D′**). (**F**) Counts of eGFP-positive neutrophils in the fins at 48hpf of *hai1a^hi2217*; *ikbkg^+/sq304* and *hai1a^hi2217*; *ikbkg^sq304*. Embryos were hemizygous for *Tg(mpx:eGFP)^i114*. n = 9; t-test; ***p<0.001. (**G–I**) Lateral DIC images of the trunk of *hai1a^+/hi2217*; *ikbkg^+/sq304* (**G**), *hai1a^hi2217*; *ikbkg^+/sq304* (**H**), and *hai1a^hi2217*; *ikbkg^sq304* (**I**). Loss of IKBKG does not rescue epidermal defects of *hai1a* mutants (arrowheads). (**J, K**) Lateral confocal projections of *Tg(6xHsa.NFKB:EGFP)^nc1* embryos reporting NfkB signalling levels at 32hpf of *hai1a^hi2217* uninjected (**J**) or injected with *duox* MO (**K**). Loss of H₂O₂ reduces NfkB signalling levels in *hai1a* mutants. Scale bars: (**A, K**) = 200 μm; (**D′, I**) = 50 μm.

The online version of this article includes the following figure supplement(s) for figure 4:

**Figure supplement 1.** NfkB signalling is elevated in *hai1a* mutants, and mutation of *ikbkg* rescues neutrophil inflammation.

was a robust increase in fluorescence, indicating that PMA activates the NfkB pathway (*Figure 6T, U*).

The phenocopy and the rescue of *hai1a* by PMA and Gq inhibition respectively imply that DAG is elevated in *hai1a* mutants. Elevated cellular DAG leads to relocalisation of Protein Kinase C isoforms to the plasma and nuclear lipid membranes where they bind DAG and become activated. Using a

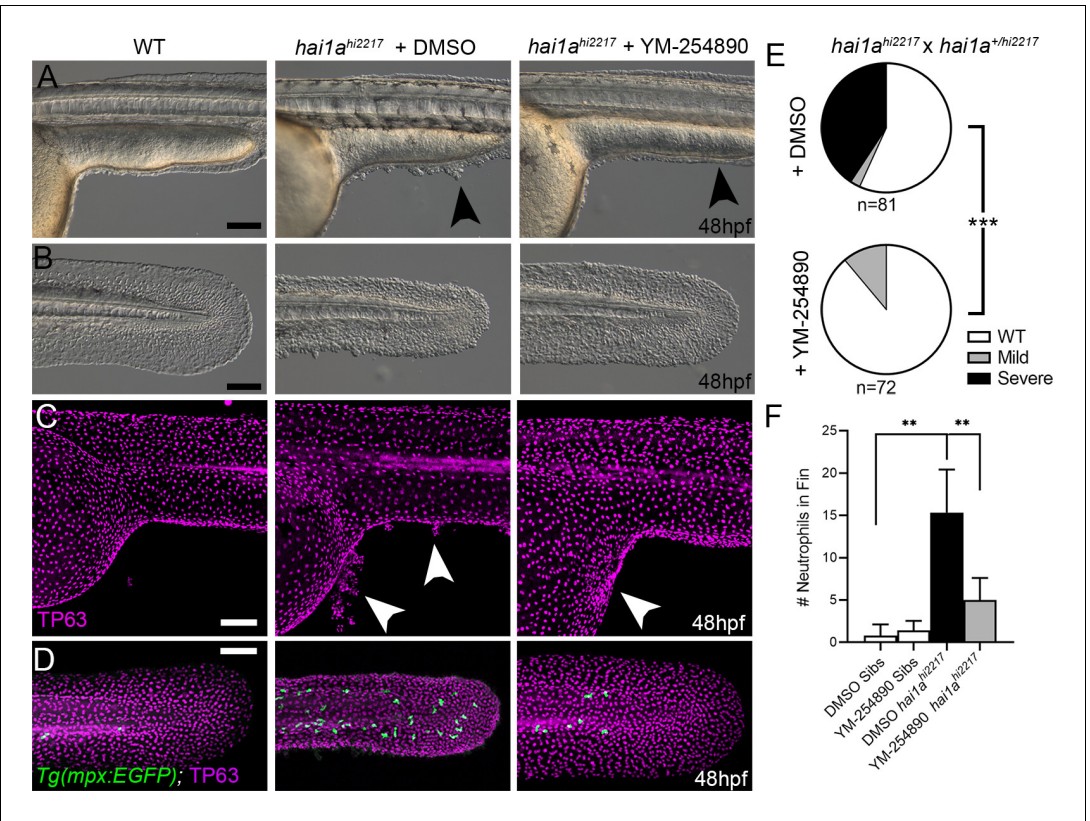

**Figure 5.** Gq inhibition rescues both epidermal and inflammation phenotypes of *hai1a* mutants. (A–D) Lateral images of ventral trunk and tail at 48hpf for WT (left panels), *hai1a^hi2217^* treated with 0.5% DMSO (middle panels), and *hai1a^hi2217^* treated with 32 μM YM-254890 (right panels). DIC micrographs are shown in (A, B), whilst projected confocal images are shown in (C, D), where embryos are immunostained for TP63 (C, D; magenta) and eGFP (D; green). Embryos in (D) are hemizygous for *Tg(mpx:eGFP)^i114^*. Arrowheads indicate region of aggregate formation lost upon treatment with Gq inhibitor YM-254890. (E) Pie charts showing proportion of embryos with no (WT; white), mild (grey) or severe (black) *hai1a* mutant epidermal phenotypes. Embryos were derived from *hai1a^hi2217/hi2217^* × *hai1a^+/hi2217^* crosses and assayed at 48hpf. Clutches treated with 0.5% DMSO (upper pie chart) were compared to those treated with 32 μM YM-254890 (lower pie chart) by Chi-squared analysis. ***p<0.001; n = 72. (F) Graph of counts of eGFP-positive neutrophils in the fins at 48hpf of *Tg(mpx:eGFP)^i114^*, or *hai1a^hi2217^; Tg(mpx: eGFP)^i114^* treated with 0.5% DMSO, or 32 μM YM-254890. n = 6; Mann–Whitney test; **p<0.01. Scale bars: (A–D) = 100 μm.

GFP-tagged PKCδ fusion protein (*Sivak et al., 2005*), we showed that in the WT embryo there was largely diffuse cytoplasmic PKCδ-GFP signal, however, it translocated to plasma and nuclear membranes in *hai1a* mutants, indicating increased levels of DAG (*Figure 7A, B*, *Figure 7—figure supplement 1A, B*). This is indeed relevant to the epidermal defects, as treatment of *hai1a^hi2217^* embryos with the PKC inhibitor, GFX109203, reduced the epidermal aggregates and disruption of fin morphology as imaged by DIC or immunostaining for TP63 (*Figure 7C–H*). Neutrophil inflammation in the epidermis was somewhat reduced, but not to a significant degree (*Figure 7E–I*). Thus, these experiments strongly suggest that epithelial defects of *hai1a* are due to DAG generation and PKC activation.

## Elevated MAPK signalling generates epithelial defects in *hai1a*

We next sought to determine which pathways downstream of PKC are responsible for the epidermal defects. The MAPK pathway is a major target pathway of multiple PKC isoforms, and activation of this pathway in zebrafish epidermis has previously been shown to induce papilloma formation which have very similar attributes to *hai1a* mutant aggregates (*Chou et al., 2015*). Although whole embryo western analysis of *hai1a* mutants failed to show an overall increase in pERK (*Armistead et al.,*

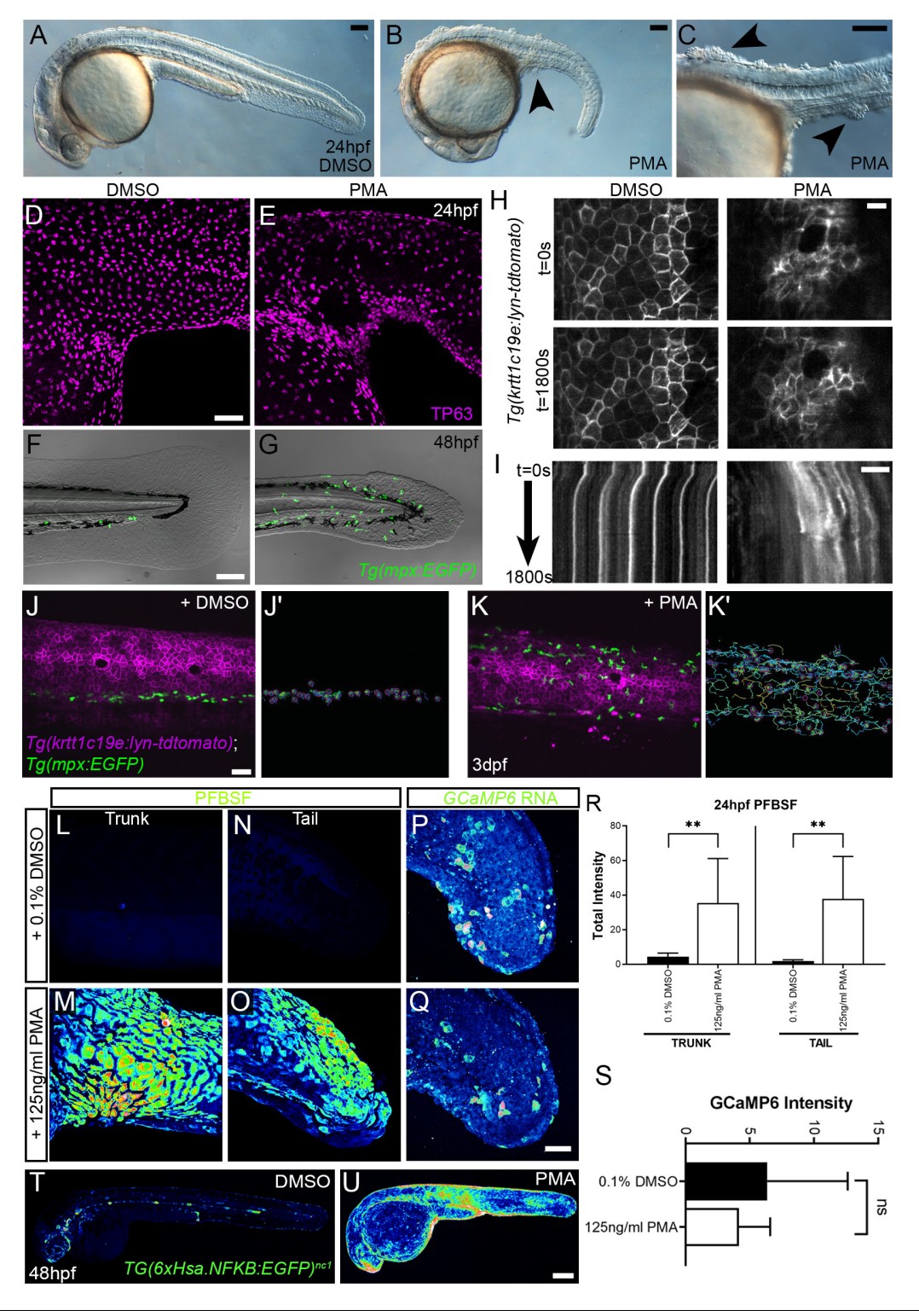

**Figure 6.** Phorbol 12-myristate 13-acetate (PMA) induces epidermal aggregates, motility, H₂O₂, NfkB, and inflammation. (**A, B**) Lateral micrographs of embryos treated with DMSO (**A**) or 125 ng/ml PMA (**B, C**) showing generation of epidermal aggregates (arrowheads). (**D, E**) Projected confocal images of the trunk of 24hpf WT embryos treated with 0.1% DMSO (**D**) or 125 ng/ml PMA (**E**) and immunostained for TP63 (magenta), showing aggregation of TP63-positive cells. (**F, G**) Projected confocal images superimposed on DIC image of the tail of 48hpf *Tg(mpx:eGFP)^{i114}* embryos treated with 0.1% DMSO (**F**) or 125 ng/ml PMA (**G**) showing fin defect and

*Figure 6 continued on next page*

*Figure 6 continued*

activation of eGFP-positive neutrophils (green, **G**). (**H, I**) Single timepoint images at t = 0 (top panels, **H**) and t = 1800 s (lower panels, **H**) and kymographs (**I**) derived from light-sheet movies (***Video 7***) of the epidermis of 3dpf *Tg(krtt1c19e:lyn-tdtomato)^sq16* larvae treated with 0.1% DMSO (left panels) or 37.5 ng/ml PMA (right panels) showing the lack of membrane stability following PMA treatment. (**J–K′**) Single frames (**J, K**) and tracks of eGFP-positive neutrophils (**J′, K′**) from light-sheet (***Video 8***) showing neutrophils labelled by eGFP and basal keratinocyte cell membranes labelled by lyn-tdTomato in the trunk of a 3dpf *Tg(krtt1c19e:lyn-tdtomato)^sq16* larva treated with 0.1% DMSO (**J, J′**) or 37.5 ng/ml PMA (**K, K′**) for 18 hr, and imaged every 20 s for 30 min. Track colour in (**J′, K′**) denotes mean velocity (dark blue 0.0 – red 0.2). (**L–O**) Projected lateral confocal views of pentafluorobenzenesulfonyl fluorescein (PFBSF) staining of 24hpf WT embryos treated with 0.1% DMSO (**L, N**) or 125 ng/ml PMA (**M, O**) showing elevation of $H_2O_2$ in the trunk (**L, M**) and tail (**N, O**). (**P, Q**) Projected confocal images of eGFP in the tail at 24hpf of WT injected with *GCaMP6s* RNA, treated with DMSO (**P**), or with 125 ng/ml PMA (**Q**). Images are temporal projections of timelapse movies taken at maximum speed intervals (2 min) and projected by time. (**R**) Plot of PFBSF fluorescent staining intensity of WT embryos treated with 0.1% DMSO or 125 ng/ml PMA in the trunk and tail. n = 6; ANOVA with Bonferroni post-test **p<0.01. (**S**) Graph comparing eGFP intensities from 24hpf *GCaMP6s* RNA timelapses in tail following treatment with DMSO and 125 ng/ml PMA. n = 10; t-test. (**T–U**) Lateral confocal projections of *Tg(6xHsa.NFKB:EGFP)^nc1* embryos reporting NfkB signalling levels at 48hpf in WT treated with DMSO (**T**) and WT treated with 125 ng/ml PMA (**U**). Scale bars: (**A, B, C, F**) = 100 µm; (**D, J, Q**) = 50 µm; (**H, I**) = 20 µm; (**U**) = 200 µm.

*2020*), we performed wholemount immunofluorescent analysis in case there was only a localised effect. Indeed, we observed a significant and localised increase in cytoplasmic pERK immunoreactivity (phospho-p44/42 MAPK (Erk1/2) (Thr$^{202}$/Tyr$^{204}$)) in the regions of epidermal aggregate formation in *hai1a* mutants and in PMA-treated embryos, including under the yolk at 24hpf and in the fins at 24hpf and 48hpf (***Figure 8A–K***, ***Figure 8—figure supplement 1A–F***). There was no increase in total ERK levels in the mutant (***Figure 8—figure supplement 1M, N***). Increased pERK was seen in both the cytoplasm and nucleus of TP63-positive cells but was only increased in the nucleus of periderm cells (***Figure 8E–E′***, ***Figure 8—figure supplement 1D***). To establish that this is an early marker of aggregate formation, and not a sequela, we stained *hai1a* mutant embryos at earlier timepoints. We found that at 16hpf regions of the epidermis have pERK staining before overt aggregation formation (***Figure 8G–H***), whilst nascent aggregates also contain pERK staining which increases in number over time (***Figure 8—figure supplement 1G–L***).

To determine if elevated pERK is causative of epidermal defects, we attempted to rescue using pERK inhibitors. Initially we used PD0325901; however, this appeared to give fin fold deformities, even in WT embryos (***Anastasaki et al., 2012***), precluding ability to assess rescue in *hai1a*, although there was a noticeable reduction in epidermal aggregates forming under the yolk-sac extension (data not shown). Instead, we tried U0126 and CI-1040, other well-known pERK inhibitors (***Allen et al., 2003***; ***Favata et al., 1998***). Both inhibitors showed a significant reduction in *hai1a* mutant epidermal aggregates under the yolk, and restoration of the overall and tail epithelial morphology, with embryos showing a *hai1a* phenotype class significantly reduced (***Figure 9A–G***, ***Figure 9—figure supplement 1A–F***). Similarly, the epidermal defects of the trunk, yolk, and tail following PMA treatment were also ameliorated by concomitant U0126 treatment (***Figure 9H, I***, ***Figure 9—figure supplement 1G, H***). Rescue of aggregates and tail morphology following PMA treatment or in *hai1a* mutants could be visualised by immunolabelling TP63 in basal keratinocyte nuclei (***Figure 9J–O***,

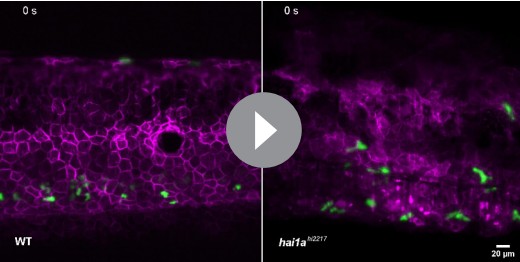

**Video 6.** Basal keratinocyte membrane and neutrophil dynamics in 3dpf wild-type and *hai1a^hi2217* larvae carrying the *Tg(krtt1c19e:lyn-tdtomato)^sq16* and *Tg(mpx:eGFP)^i114* transgenes. Projected light-sheet timelapses of the trunk of 3dpf WT (left) and *hai1a^hi2217* (right) larvae with neutrophils and basal keratinocyte membranes labelled by eGFP and lyn-tdTomato, respectively. Both larvae carried the *Tg(krtt1c19e:lyn-tdtomato)^sq16*; *Tg(mpx:eGFP)^i114* transgenes. The *hai1a* mutants have highly dynamic neutrophils and keratinocyte membrane dynamics. Scale bar: 20 µm. https://elifesciences.org/articles/66596#video6

**Video 7.** Basal keratinocyte membranes in DMSO and phorbol 12-myristate 13-acetate (PMA)-treated 3dpf *Tg (krtt1c19e:lyn-tdtomato)^sq16* larvae. Zoomed projected light-sheet timelapses of basal keratinocyte membranes labelled by lyn-tdTomato in the trunk of 3dpf *Tg(krtt1c19e:lyn-tdtomato)^sq16* larvae treated with 0.1% DMSO (left) and 37.5 ng/ml PMA (middle and right) for 18 hr. Membranes are stable in DMSO-treated larvae but were dynamic in PMA-treated larvae. Images were captured every 20 s. Scale bar: 10 μm.
https://elifesciences.org/articles/66596#video7

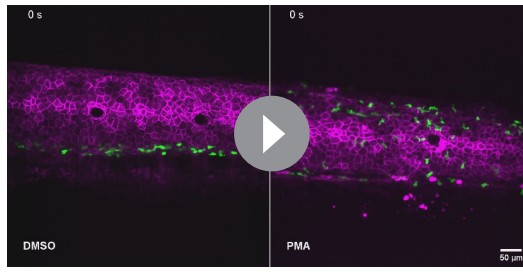

**Video 8.** Neutrophils and basal keratinocyte membranes in DMSO and phorbol 12-myristate 13-acetate (PMA)-treated 3dpf *Tg(krtt1c19e:lyn-tdtomato)^sq16*; *Tg(mpx:eGFP)^i114* larvae. Lateral projection of light-sheet timelapse of neutrophils labelled by eGFP and basal keratinocyte cell membranes labelled by lyn-tdTomato in the trunks of 3dpf *Tg(krtt1c19e:lyn-tdtomato)^sq16* larva treated with 0.1% DMSO (left) and 37.5 ng/ml PMA (right) for 18 hr. PMA treatment leads to slightly dynamic cell membranes and motile neutrophils. Images were captured every 20 s for 30 min. Scale bar: 50 μm.
https://elifesciences.org/articles/66596#video8

*Figure 9—figure supplement 1I, J*). Initiating U0126 treatment later at 26hpf led to only a partial rescue, indicating that the epidermal phenotypes were likely due to sustained pERK activation (*Figure 9—figure supplement 1K–M'*).

Treatment with U0126 did not significantly reduce neutrophil inflammation of *hai1a* mutants or PMA treatment (*Figure 9L–P*). This suggests that the inflammation phenotype is not simply a consequence of the epidermal defects. Furthermore, dye penetration assays showed that the epithelial barrier was not globally and overtly compromised in *hai1a*, underscoring that inflammation is not simply a consequence of epithelial defects (*Figure 9—figure supplement 2A–H*). It has been shown that the epidermal defects in *hai1a* are associated with loss of E-cadherin from adherens junctions (*Carney et al., 2007*). As there was a rescue of the epithelial phenotype following pERK inhibition, we looked at the status of the adherens junction marker β-catenin. Whilst the WT basal epidermal cells of the 48hpf tail showed strong staining at the membrane, *hai1a* mutants and PMA-treated embryos showed a significant loss of β-catenin at the membrane and increase in the cytoplasm (*Figure 9Q–V, Y, Z*). Treatment of *hai1a* mutants with U0126 restored the membrane localisation of β-catenin (*Figure 9W, X*, AA).

## Phosphorylation of cytoplasmic RSK by pERK leads to loss of E-cadherin at the *hai1a* keratinocyte membrane

As increased pERK appeared to contribute strongly to loss of adherens junctions and removal of E-cadherin/β-catenin from the membrane, we sought to determine how pERK signalling might affect adherens junctions. We predicted that this would occur through a cytoplasmic target of pERK as we have previously shown that there is no transcriptional downregulation of E-cadherin levels in *hai1a*, making a nuclear transcription factor target less likely to be relevant (*Carney et al., 2007*). The p90RSK family of kinases represents direct cytoplasmic targets of Erk1/2 phosphorylation which regulate cell motility, and thus were good candidates for mediators disrupting cell-cell adhesion (*Čáslavský et al., 2013*; *Tanimura and Takeda, 2017*). We determined that at least RSK2a (=p90RSK2a, encoded by *rps6ka3a*) is expressed in basal keratinocytes at 24hpf (*Figure 10A, B*). To gauge if there was an alteration in phosphorylation of RSK family members in the epidermis of *hai1a* mutants, we used an antibody which detects a phosphorylated site of mouse p90RSK (Phospho-Thr[348]). This site is phosphorylated in an ERK1/2-dependent manner (*Romeo et al., 2012*). We noticed a substantial increase in cytoplasmic signal in both *hai1a* mutants and PMA-treated embryos. Where p90RSK-pT[348] signal was largely nuclear in both basal and periderm cells in WT, it was more broadly observed in *hai1a* mutant fins, with an increase in the cytoplasm leading to a more uniform staining (*Figure 10C–D'*). This increase in cytoplasmic levels of p90RSK-pT[348] was observable at 17hpf prior to epithelial defects (*Figure 10—figure supplement 1A–C*). p90RSK cytoplasmic signal

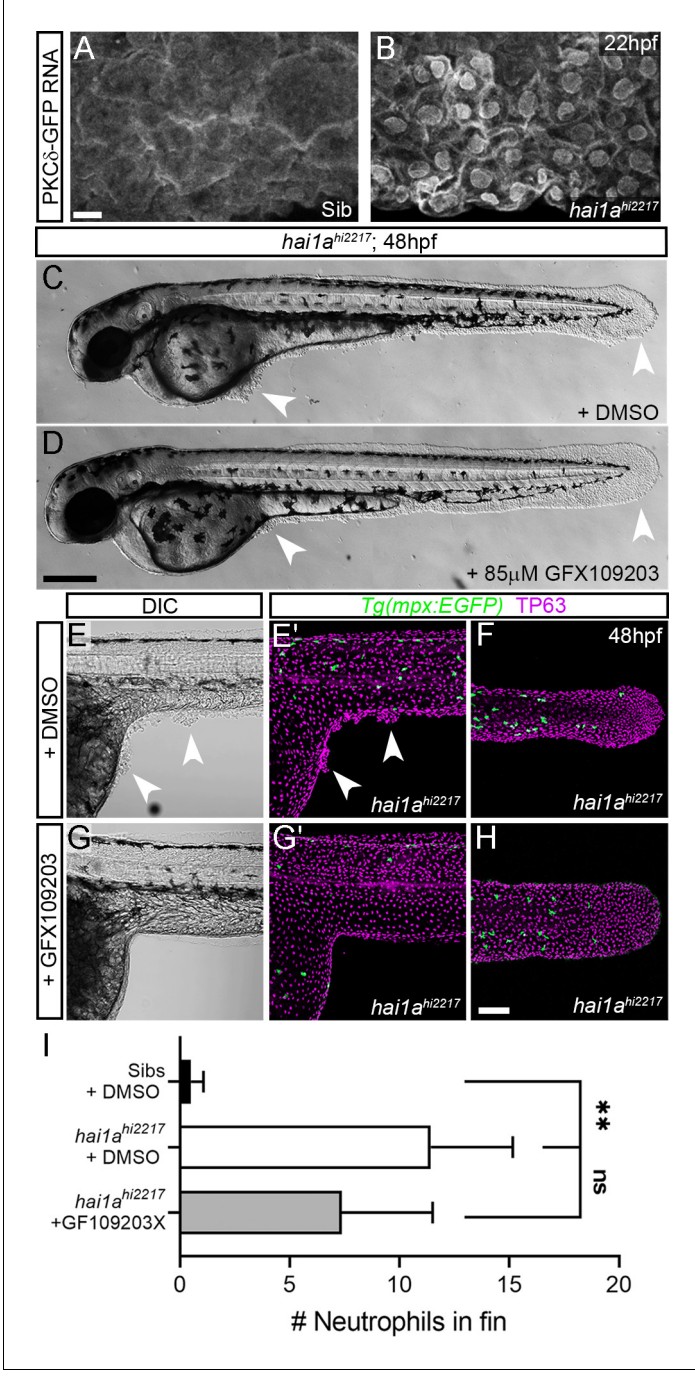

**Figure 7.** Inhibition of PKC rescues epidermal defects of *hai1a*. (A, B) Confocal images of the ventral fin of 22hpf sibling (A) or *hai1a^hi2217* (B) embryos injected with RNA encoding PKCδ-GFP. Mostly cytoplasmic distribution in sibling was relocated to cell and nuclear membranes in *hai1a* mutants. (C, D) Lateral brightfield images of 48hpf *hai1a^hi2217* larvae treated with 0.5% DMSO (C) or 85 μM GFX109203 (D). Epidermal aggregates and fin deterioration are rescued by the PKC inhibitor (arrowheads). (E–H) DIC (E, G) and projected confocal images (E', G', F, H) of *hai1a^hi2217*; *Tg(mpx:eGFP)^i114* trunk at 24hpf (E–E', G–G') and tail at 48hpf (F, H), either treated with 0.5% DMSO (E–F) or 85 μM GFX109203 (G–H). Embryos are immunostained for TP63 (magenta) and eGFP (green), highlighting rescue of epidermal phenotype and partial rescue of neutrophils by GFX109203. (I) Counts of eGFP-positive neutrophils in the fins at 48hpf of *Tg(mpx:eGFP)^i114*, or *hai1a^hi2217*; *Tg(mpx:eGFP)^i114* treated with 0.5% DMSO or 85 μM GFX109203. n = 8; ANOVA, Dunn's multiple comparisons; **p<0.01. Scale bars: (A) = 10 μm; (D) = 200 μm; (H) = 100 μm.

The online version of this article includes the following figure supplement(s) for figure 7:

*Figure 7 continued on next page*

*Figure 7 continued*

**Figure supplement 1.** Relocation of PKCδ-GFP to membranes in *hai1a* mutants.

was lost upon U0126 and GFX109203 treatments, showing that it was pERK and PKC dependant (*Figure 10E, E′*, *Figure 10—figure supplement 1D, E*). Similarly, increased cytoplasmic p90RSK-pT$^{348}$ was observed following PMA treatment which was reduced by co-treatment with U0126 (*Figure 10F–H′*). The increase in cytoplasmic p90RSK-pT$^{348}$ signal, and its reduction by U0126, was significant in both *hai1a* mutants and PMA-treated embryos (*Figure 10I, J*).

If phosphorylation of an RSK protein is required for mediating the pERK epidermal defects in *hai1a* mutants, then inhibition of RSK should rescue the epidermal defects. As morpholino-targeted inhibition of *rps6ka3a* was unsuccessful, we employed established pan-RSK inhibitors BI-D1870 and dimethyl fumarate (*Andersen et al., 2018*; *Sapkota et al., 2007*). Dimethyl fumarate treatment reduced the extent of cytoplasmic p90RSK-pT$^{348}$ in *hai1a* (*Figure 10—figure supplement 1F, G*). We noted that both inhibitors were able to reduce epidermal aggregates in *hai1a* mutants and restore fin morphology when visualised by DIC or TP63 immunofluorescence (*Figure 10K–N*, *Figure 10—figure supplement 1H, I, K, L*). Reduction of mutant phenotype classes was significant at both 24hpf and 48hpf (*Figure 10—figure supplement 1J*). We then assayed if RSK inhibition can reduce the aberrant cytoplasmic E-cadherin staining in *hai1a* mutant basal keratinocytes and observed that dimethyl fumarate treatment restored membrane localisation of E-cadherin in the mutants (*Figure 10O–Q′*). Thus, phosphorylation of RSK proteins is altered in *hai1a* mutants, and their inhibition appears to restore E-cadherin to the membrane and reduce epidermal aggregate formation.

## Discussion

There are a number of similarities between loss of Hai1a in zebrafish and overexpression of Matriptase in the mouse epidermis, including inflammation, hyperproliferation, and enhanced keratinocyte motility, suggesting conservation of downstream pathways. What the conserved ancestral role of the Matriptase-Hai1 might have been is unclear. Matriptase dysregulation in the mouse is associated with cancer progression (*Martin and List, 2019*). Tumours have long been considered to represent non-healing wounds, and the cellular- and tissue-level phenotypes of *hai1a* have similarities to tumours. Epidermal cells in zebrafish transformed by MAPK activation both promote and respond to inflammation through similar mechanisms to wound responses (*Feng et al., 2010*; *Schäfer and Werner, 2008*). Further, tissue damage of the zebrafish epidermis perturbs osmolarity and releases nucleotides, leading to inflammation and epithelial cell motility, with the resulting phenotypes strikingly similar to *hai1a* mutants (*de Oliveira et al., 2014*; *Enyedi and Niethammer, 2015*; *Gault et al., 2014*; *Hatzold et al., 2016*). Indeed, the tissue responses initiated by loss of zebrafish Hai1a have been previously suggested to represent an early injury response (*Schepis et al., 2018*), whilst PAR2 synergises with P2Y purinergic and EGF receptors to promote cell migration in scratch assays (*Shi et al., 2013*). Thus our analysis supports the previous hypothesis of the Hai1-Matriptase system as a component of tissue injury responses (*Schepis et al., 2018*), which, if inappropriately activated, promotes carcinoma.

The various molecular pathways known to be activated by Matriptase have not been fully delineated or integrated. Par2 has previously been shown to be required for the *hai1a* phenotype in zebrafish and contributes to the phenotypes of Matriptase overexpression in the mouse. Exactly which heterotrimeric G-protein Par2 is activating in vivo and how this links to phenotypes has not been identified. Our analyses allow us to propose a pathway downstream of Par2 which accounts for both the inflammatory and the epidermal phenotypes (*Figure 11*). Firstly, inhibition of Gq rescued both the inflammation and epithelial defects. PAR2 activation of Gq has been documented to occur in many cell types including keratinocytes, where inhibition of Gq and PKC reduces PAR2-mediated Nfκb signalling (*Böhm et al., 1996*; *Goon Goh et al., 2008*; *Macfarlane et al., 2005*). Although we were unable to rescue *hai1a* phenotypes with a PLC inhibitor due to toxicity, genetic sensors demonstrated increased levels of Ca$^{++}$ and DAG in *hai1a* epidermis. Our analysis demonstrated that the different products of PIP2 hydrolysis appear to invoke the two main *hai1a* phenotypes to different

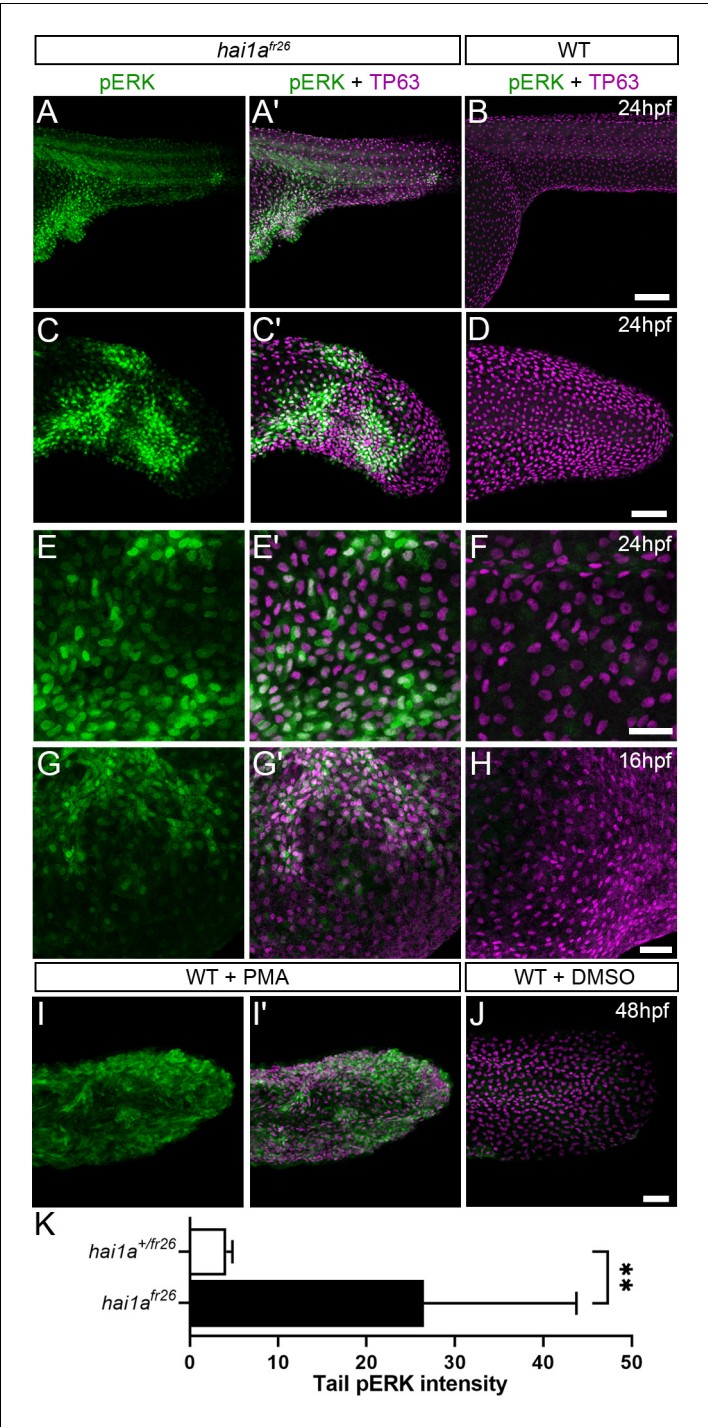

**Figure 8.** Elevation of pERK levels in phorbol 12-myristate 13-acetate (PMA)-treated and *hai1a* mutant epidermis.
(A–L) Lateral projected confocal images of trunks (A, A', B, E, E', F), yolk surface (G, G', H) and tails (C, C', D, I, I',
J) of embryos immunostained for TP63 (A', B, C', D, E', F, G', H, I', J; magenta) and pERK (A, J; green) at 24hpf (A–
F), 16hpf (G–H), and 48hpf (I–J). Both *hai1a^{fr26}* (A, A', C, C', E, E', G, G') and 125 ng/ml PMA-treated (I, I') embryos
show increased epidermal pERK levels compared to untreated WT (B, D, F, H, J). Elevation of epidermal pERK in
*hai1a^{fr26}* mutants and PMA- treated embryos is seen in the trunk (A, E) and tail (C, I) as well as in epidermis over
the yolk prior to overt phenotype manifestation (G). (K) Quantification of pERK immunofluorescent intensity in the
tail of 24hpf *hai1a^{fr26}* larvae compared to siblings. n = 5; Mann–Whitney test; **p<0.01. Scale bars: (B) = 100 µm;
(D, H, J) = 50 µm; (F) = 20 µm.
The online version of this article includes the following figure supplement(s) for figure 8:

*Figure 8 continued on next page*

*Figure 8 continued*

**Figure supplement 1.** Elevation of pERK levels in phorbol 12-myristate 13-acetate (PMA)-treated and *hai1a* mutant epidermis.

extents. IP$_3$R-dependent calcium release in *hai1a* epidermis was required for Duox activity, high hydrogen peroxide levels, and, later, increased NfkB signalling. Reduction of these attenuated the inflammatory, but not epithelial, defects. Conversely, inhibiting the DAG receptor, PKC, rescued the epithelial phenotypes, and the inflammation slightly. The DAG analogue, PMA, phenocopied the epidermal defects of *hai1a* mutants but also increased H$_2$O$_2$, NfkB, and neutrophil inflammation, indicating that PKC activation may be sufficient, but not necessary, for inflammation. This is in line with known activation of Duox and IKK by PKC (*Rigutto et al., 2009*; *Turvey et al., 2014*). In addition, expression of activated Ras in zebrafish keratinocytes has been shown to lead to H$_2$O$_2$ release and neutrophil attraction (*Feng et al., 2010*). Thus, there is likely to be dual contribution to the inflammatory phenotype from IP3 and DAG. It is important to stress however that the inflammation is not simply a result of epithelial defects or an overt loss of barrier. Firstly, we see increase in Ca$^{++}$ and H$_2$O$_2$ very early in the epidermis prior to skin defects. Secondly, barrier assays failed to conclusively show a broad increase in permeability. Finally, rescue of epithelial defects by PKC and pERK inhibition did not fully rescue the inflammation. We conclude in our model that DAG contributes to both aspects of the phenotype, but IP$_3$ promotes only the inflammation.

Seminal experiments in transgenic mice overexpressing Matriptase in the epidermis and treated with a DMBA/PMA regime concluded that Matriptase and PMA activate functionally similar carcinoma promoting pathways (*List et al., 2005*). Our subsequent analysis suggests that this would include the MAPK pathway as we see increased phosphorylated-ERK in the epidermis of both *hai1a* mutants and also PMA-treated embryos. That we can rescue the epithelial defects using a MEK inhibitor indicated that this increase in epidermal pERK is likely critical to the phenotype. The MAPK pathway is known to regulate cell motility (*Tanimura and Takeda, 2017*). In the zebrafish epidermis, misexpression of activated MEK2 generated papillomas with remarkable resemblance to the epidermal aggregates in *hai1a* mutants (*Chou et al., 2015*), and which are not overtly proliferative. In astrocytes and oesophageal or breast tumour cell lines, PAR2 stimulates migration and invasiveness through MAPK/ERK, activation of which required Gq and PIP2 hydrolysis (*Jiang et al., 2004*; *McCoy et al., 2010*; *Morris et al., 2006*; *Sheng et al., 2019*).

One of the main molecular defects defined for zebrafish *hai1a* is the removal of adherens junction proteins from the membrane (*Carney et al., 2007*). MAPK signalling has been shown to reduce E-cadherin expression at adherens junctions and promote cytoplasmic accumulation through phosphorylation of the effector, RSK (*Čáslavský et al., 2013*). Like Matriptase, activation of RSK2 is associated with tumour progression, promoting invasiveness and metastasis of glioblastomas and head and neck squamous cell carcinomas (*Kang et al., 2010*; *Sulzmaier et al., 2016*). Promotion of invasiveness has also been noted for activated RSK1, which promotes invasion of melanoma clinically as well as in vitro and zebrafish melanoma models (*Salhi et al., 2015*). Intriguingly, proximity protein labelling has identified p120-catenin as a target of RSK phosphorylation. This catenin promotes cell-cell adhesion by stabilising cadherins at junctions, a function inhibited by RSK phosphorylation (*Méant et al., 2020*). More broadly, RSK2 activity promotes cell motility through other mechanisms, including inactivation of Integrins and activation of the RhoGEF, LARG (*Gawecka et al., 2012*; *Shi et al., 2018*). Thus, we propose that pERK signalling, through RSK members, significantly contributes to dissolution of adherens junctions and the *hai1a* epidermal phenotype. We observed increased pERK in the cytoplasm and also the nucleus of keratinocytes, with comparatively more nuclear levels in periderm cells. Thus, whilst RSKs are phosphorylated by pERK, it is also likely that other cytoplasmic and also nuclear targets, such as cFos and Ets transcription factors, may also be activated, and that there are underlying transcriptional changes in *hai1a* mutants. It is not clear why pERK shows slightly different subcellular localisation patterns between the two different epidermal layers, but the two layers do respond differently to ErbB2 inhibition (*Schepis et al., 2018*), whilst calcium is recently described to alter nuclear shuttling of pERK (*Chuderland et al., 2020*).

Our model for how Matriptase invokes cellular responses is highly likely to be incomplete. Indeed, others have indicated MMPs, HB-EGF, EGFR, and AKT and are downstream of Matriptase and PAR2

function (*List et al., 2005*; *Schepis et al., 2018*; *Darmoul et al., 2004*; *Chung et al., 2013*; *Rattenholl et al., 2007*). Furthermore, Matriptase promotes HGF–cMet signalling in mouse (*Szabo et al., 2011*). We do not think that these conflict with our model but will interface with it. A number of reports have demonstrated that PI3K/AKT and MEK/ERK function in parallel downstream of PAR2 (*Sheng et al., 2019*; *Tanaka et al., 2008*; *van der Merwe et al., 2009*). Furthermore, there is evidence that PKC activates both MEK/ERK and EGFR independently following PAR2 stimulation, and that PI3K is activated by PAR2 via Gq (*Wang and DeFea, 2006*; *Al-Ani et al., 2010*). Cell identity, subcellular localisation, β-arrestin scaffolding, and biased agonism/antagonism are known to generate alternative downstream outputs from PAR2 (*Zhao et al., 2014*). To understand fully the roles of Matriptase and PAR2 in epithelial homeostasis and carcinoma, it will be critical to map how, when, and where they activate different downstream pathways.

# Materials and methods

## Key resources table

| Reagent type (species) or resource | Designation | Source or reference | Identifiers | Additional information |
|---|---|---|---|---|
| Gene (*Danio rerio*) | *hai1a* | GenBank | NM_213152 | =*spint1a* |
| Gene (*Danio rerio*) | *matriptase1a* | GenBank | NM_001040351 | =*st14* a |
| Gene (*Danio rerio*) | *duox* | GenBank | XM_017354273 | =*dual oxidase* |
| Gene (*Danio rerio*) | *ikbkg* | GenBank | NM_001014344 | =*ikky* <br> =*nemo* |
| Gene (*Danio rerio*) | *nfkbiaa* | GenBank | NM_213184 | =*ikbaa* |
| Gene (*Danio rerio*) | *rps6ka3a* | GenBank | NM_212786 | =*RSK2a* <br> =*p90RSK2a* |
| Gene (*Danio rerio*) | *tp63* | GenBank | NM_152986 | =*delta* Np63 |
| Strain, strain background (*Escherichia coli*) | Top10 | Invitrogen | C404010 | Chemical competent cells |
| Strain, strain background (*Danio rerio*) | AB | ZIRC | | Wild-type strain |
| Strain, strain background (*Danio rerio*) | TL | ZIRC | | Wild-type strain |
| Genetic reagent (*Danio rerio*) | Tg(mpx:EGFP)[i114] | Uni of Sheffield PMID:16926288 | ZFIN ID: ZDB-ALT-070118-2 | |
| Genetic reagent (*Danio rerio*) | Tg(fli1:EGFP)[y1] | ZIRC PMID:16671106 | ZFIN ID: ZDB-ALT-011017-8 | |
| Genetic reagent (*Danio rerio*) | hai1a[fr26] | Hammerschmidt lab; Max Planck Freiburg PMID:31819976 | ZFIN ID: ZDB-ALT-200618-2 | =*spint1a*[fr26] |
| Genetic reagent (*Danio rerio*) | hai1a[hi2217] | Nancy Hopkins lab; Massachusetts Institute of Technology PMID:17728346 | ZFIN ID: ZDB-ALT-040924-4 | =*spint1a*[hi2217Tg] |
| Genetic reagent (*Danio rerio*) | ddf[ti251] | Nuesslein-Volhard lab; Max Planck Tuebingen PMID:9007245 | ZFIN ID: ZDB-ALT-980203-1462 | =*dandruff* spint1a[ti251] ==*hai1a*[ti251] |
| Genetic reagent (*Danio rerio*) | ddf[t419] | Nuesslein-Volhard lab; Max Planck Tuebingen PMID:9007245 | | =*dandruff* spint1a[t419] ==*hai1a*[t419] |
| Genetic reagent (*Danio rerio*) | st14a[sq10] | Our lab PMID:31645615 | ZFIN ID: ZDB-ALT-200219-5 | |
| Genetic reagent (*Danio rerio*) | Tg(6xNFkB:EGFP)[nc1] | Rawls lab PMID:21439961 | ZFIN ID: ZDB-ALT-120409-6 | |

*Continued on next page*

*Continued*

| Reagent type (species) or resource | Designation | Source or reference | Identifiers | Additional information |
|---|---|---|---|---|
| Genetic reagent (*Danio rerio*) | *Tg(krtt1c19e:LY-Tomato)*[sq16] | Our lab. Lee et al: PMID:24400120 | ZFIN ID: ZDB-ALT-140424-2 | |
| Genetic reagent (*Danio rerio*) | *Tg(actb2:GCaMP6s, myl7:mCherry)*[lkc2] | This paper | | Plasmid from Solnica-Krezel Lab. Injected with Tol2 RNA to make line |
| Antibody | Chicken anti-eGFP antibody | Abcam | ab13970, RRID:AB_300798 | 1:500 |
| Antibody | Rabbit anti-eGFP | Torrey Pines Biolabs | Tp401 RRID:AB_10013661 | 1:500 |
| Antibody | Rabbit anti-FITC | Thermo Fisher Scientific | 71-1900 RRID:AB_2533978 | 1:200 |
| Antibody | Rabbit anti-p90RSK (Phospho-Thr[348]) | GenScript | A00487 | 1:100 |
| Antibody | Rabbit anti-beta catenin | Abcam | ab6302 RRID:AB_305407 | 1:200 |
| Antibody | Mouse anti-E-cadherin | BD Biosciences | 610181 RRID:AB_397580 | 1:200 |
| Antibody | Mouse anti-Tp63 | Biocare Medical | CM163 RRID:AB_10582730 | 1:200 |
| Antibody | Rabbit anti-phospho-p44/42 MAPK (Erk1/2) (Thr[202]/Tyr[204]) | Cell Signaling Technology | Cat# 4370, RRID:AB_2315112 | 1:100 |
| Antibody | Rabbit anti-p44/42 MAPK (Erk1/2) | Cell Signaling Technology | Cat# 9102, RRID:AB_330744 | 1:100 |
| Antibody | Alexa Fluor-488 Donkey anti-rabbit | Life Technologies | A21206 RRID:AB_2535792 | 1:700 |
| Antibody | Alexa Fluor-647 Donkey anti-rabbit | Life Technologies | A31573 RRID:AB_253618 | 1:700 |
| Antibody | Alexa Fluor-546 Donkey anti-mouse | Life Technologies | A10036 RRID:AB_2534012 | 1:700 |
| Antibody | Alexa Fluor-488 Goat anti-chicken | Life Technologies | A11039 RRID:AB_2534096 | 1:700 |
| Recombinant DNA reagent | pCS2+-PKCδ-GFP | Amaya Lab, Uni of Manchester PMID:15866160 | | For making *PKCd-GFP* RNA |
| Recombinant DNA reagent | pT3Ts-Tol2 | Ekker Lab, Mayo Clinic PMID:17096595 | Addgene Plasmid #31831 RRID:Addgene_31831 | |
| Recombinant DNA reagent | pCS2+-GCaMP6s | Solnica-Krezel Lab, Washington University School of Medicine, St. Louis, MO | | For making *GCaMP6s* RNA |
| Recombinant DNA reagent | p(actb2:GCaMP6s, myl7:mCherry) | Solnica-Krezel Lab, Washington University School of Medicine, St. Louis, MO. PMID:28322738 | | For making stable transgenic line |
| Sequence-based reagent | *duox* morpholino | GeneTools | PMID:19494811 | 5' AGTGAATTAGAGAAA TGCACCTTTT 3' (0.4 mM) |
| Sequence-based reagent | *p53* morpholino | GeneTools | PMID:19494811 | 5' GCGCCATTGCTTTGCA AGAATTG 3' (0.2 mM) |
| Sequence-based reagent | Oligo(dT)12–18 Primer | Invitrogen | PMID:18418012 | |
| Sequence-based reagent | *nfkbiaa* | This paper | PCR primers | F-5' AGACGCAAAGGAGC AGTGTAG 3' R- 5' TGTGTGTCTGCCGA AGGTC 3' |

*Continued on next page*

*Continued*

| Reagent type (species) or resource | Designation | Source or reference | Identifiers | Additional information |
|---|---|---|---|---|
| Sequence-based reagent | *eef1a1l1* | This paper | PCR primers | F'-5' CTGGAGGCCAGC TCAAACAT 3' R-5' ATCAAGAAGAGTAGT ACCGCTAGCATTAC 3' |
| Sequence-based reagent | *rps6ka3a* in situ probe | This paper | PCR primers for cloning probe | F'-5' ATACTCCAGTCCC ACCGGA 3' R- 5'TGGTGATGATGGT AGACTCGC 3' |
| Peptide, recombinant protein | Proteinase K | Thermo Scientific | EO0491 | 0.5 µg/µl |
| Commercial assay or kit | SuperScript III Reverse Transcriptase | Invitrogen | 18080093 | |
| Commercial assay or kit | TRIzol Reagent | Invitrogen | 15596026 | |
| Commercial assay or kit | GoTaq G2 Green Master Mix | Promega | M7823 | Functions used: TrackMate Reslice Average Intensity |
| Commercial assay or kit | iTaq Universal SYBR Green Supermix | Bio-Rad | 1725121 | Functions used: Spot |
| Commercial assay or kit | mMESSAGE mMACHINE SP6 Transcription Kit | Invitrogen | AM1340 | Tests: Student's t-test, Chi-squared test, Mann–Whitney test, ANOVA with Bonferroni or Dunn's post-tests |
| Commercial assay or kit | mMESSAGE mMACHINE T3 Transcription Kit | Invitrogen | AM1348 | |
| Commercial assay or kit | MEGAshortscript T7 Transcription Kit | Invitrogen | AM1350 | |
| Commercial assay or kit | pGEM-T Easy | Promega | A137A | |
| Commercial assay or kit | pCR 2.1-TOPO TA vector | Invitrogen | K450040 | |
| Commercial assay or kit | QIAquick PCR Purification Kit | Qiagen | 28104 | |
| Commercial assay or kit | DIG RNA Labeling Kit | Roche | 11175025910 | |
| Commercial assay or kit | SP6 RNA Polymerase | Roche | 10 810 274 001 | |
| Commercial assay or kit | NBT/BCIP Stock Solution | Roche | 11681451001 | |
| Chemical compound, drug | Diphenyleneiodonium chloride | Sigma-Aldrich | D2926 | 40 µM |
| Chemical compound, drug | Thapsigargin | Sigma-Aldrich | T9033 | 6.25 µM |
| Chemical compound, drug | Bisindolylmaleimide I (GF109203X) | Selleckchem | S7208 | 85 µM |
| Chemical compound, drug | YM-254890 | FocusBiomolecules | 10-1590-0100 | 32 µM |
| Chemical compound, drug | 2-Aminoethyl diphenylborinate | Sigma-Aldrich | D9754 | 2.5 µM |
| Chemical compound, drug | BI-D1870 | Axon Medchem | Axon-1528 | 1.2 µM |

*Continued*

| Reagent type (species) or resource | Designation | Source or reference | Identifiers | Additional information |
|---|---|---|---|---|
| Chemical compound, drug | Dimethyl fumarate | Sigma-Aldrich | 242926 | 9 µM |
| Chemical compound, drug | Phorbol 12-myristate 13-acetate | Sigma-Aldrich | P8139 | 37.5 or 125 ng/ml |
| Chemical compound, drug | U0126 | Cell Signaling Technology | 9903 | 100 µM |
| Chemical compound, drug | PD184352 (CI-1040) | Selleckchem | S1020 | 1.3 µM |
| Chemical compound, drug | DAPI (4′,6-diamidino-2-phenylindole, dihydrochloride) | Invitrogen | D1306 | 5 µg/ml |
| Chemical compound, drug | Penta-fluorobenzenesulfonyl fluorescein | Cayman Chemicals | 10005983 | 12.5 µM |
| Chemical compound, drug | *Fluorescein isothiocyanate–dextran* | Sigma-Aldrich | FD4 | 2.5 mg/ml |
| Software, algorithm | Fiji (ImageJ 1.52p) | NIH | https://imagej.nih.gov/ | Functions used: TrackMate Reslice Average Intensity |
| Software, algorithm | Imaris 9.6.0 | Oxford Instruments | | Functions used: Spot |
| Software, algorithm | Prism 9.1.1 | GraphPad | | Tests: Student's t-test, Chi-squared test, Mann–Whitney test, ANOVA with Bonferroni or Dunn's post-tests |
| Software, algorithm | Photoshop 22.1.1 release | Adobe | | |

## Zebrafish husbandry and lines

Fish were housed at the IMCB and the NTU zebrafish facilities under IACUC numbers #140924 and #A18002, respectively, and according to the guidelines of the National Advisory Committee for Laboratory Animal Research. Embryos were derived by natural crosses and staged as per *Kimmel et al., 1995* and raised in 0.5× E2 medium (7.5 mM NaCl, 0.25 mM KCl, 0.5 mM $MgSO_4$, 75 µM $KH_2PO_4$, 25 µM $Na_2HPO_4$, 0.5 M $CaCl_2$, 0.35 mM $NaHCO_3$). Anaesthesia was administered in E2 medium (embryos) or fish tank water (adults) using 0.02% pH 7.0 buffered Tricaine MS-222 (Sigma). The *hai1a/ddf* alleles used were *hai1a*[hi2217], *hai1a*[fr26], *ddf*[ti251], and *ddf*[t419]. The *st14a*[sq10] allele was generated previously (*Lin et al., 2019*). For imaging neutrophils and keratinocytes, the transgenic lines *Tg(mpx:EGFP)*[i114] (*Renshaw et al., 2006*) and *Tg(krtt1c19e:lyn-tdtomato)*[sq16] (*Lee et al., 2014*) were used, whilst early leukocytes were imaged with *Tg(fli1:EGFP)*[y1] (*Redd et al., 2006*). To image NfkB pathway activity, the *Tg(6xHsa.NFKB:EGFP)*[nc1] sensor line was used (*Kanther et al., 2011*). Calcium imaging was performed by injection of *GCaMP6s* RNA (see below) or using a *Tg(actb2:GCaMP6s, myl7:mCherry)*[lkc2] stable transgenic line, generated via plasmid (*Chen et al., 2017*) and *Tol2* RNA co-injection.

## Genomic DNA and RNA extraction, reverse transcription, and PCR

Adult fin clips or embryos were isolated following anaesthesia, and genomic DNA extracted by incubation at 55°C for 4 hr in Lysis buffer (10 mM Tris pH 8.3, 50 mM KCl, 0.3% Tween20, 0.3% Nonidet P-40, 0.5 µg/µl Proteinase K). PCRs were performed using GoTaq (Promega) on a Veriti thermal cycler (Applied Biosystems) and purified with a PCR purification kit (Qiagen). TRIzol (Invitrogen) was used for RNA extraction following provided protocol, and cDNA generated from 1 µg total RNA

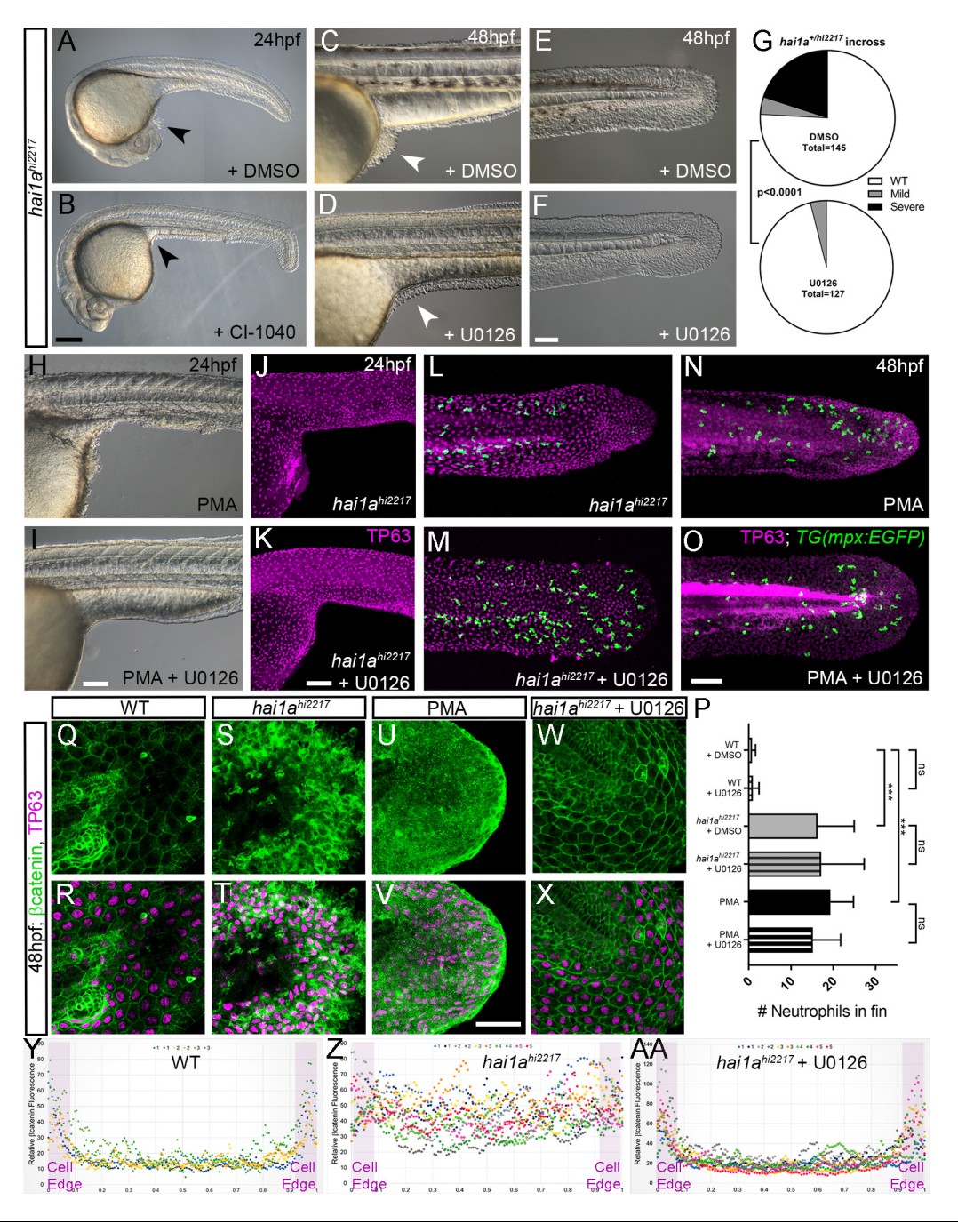

**Figure 9.** Rescue of the *hai1a* epidermal phenotype by pERK inhibitors. (A–F) Lateral DIC images of 24hpf (A, B) or 48hpf (C–F) *hai1a^hi2217* embryos treated with either DMSO (A, C, E), 1.3 μM CI-1040 (B), or 100 μM U0126 (D, F) showing rescue of general morphology (B), trunk (D), and tail (F) epidermal phenotypes compared to DMSO-treated *hai1a^hi2217*. Epidermal aggregates under the yolk are reduced in the treated mutants (A–D; arrowheads). (G) Proportion of 48hpf larvae derived from *hai1a^+/hi2217* incross showing mild or severe *hai1a* epidermal phenotype following DMSO (upper) or U0126 (lower) treatment (Chi-squared test). (H, I) Lateral DIC images of 24hpf embryo treated with 125 ng/ml phorbol 12-myristate 13-acetate (PMA) (H) or PMA and U0126 (I). Yolk-associated epidermal aggregates are reduced. (J–M) Lateral projected confocal images of *hai1a^hi2217*; *Tg(mpx: eGFP)^i114* trunk at 24hpf (J, K) and tail at 48hpf (L, M), either treated with 0.5% DMSO (J, L) or U0126 (K, M). Embryos are immunostained for TP63 (magenta) and eGFP (green), highlighting rescue of epidermal phenotype but no reduction of neutrophils. (N, O) Lateral projected confocal images of *Tg(mpx:eGFP)^i114* treated with PMA alone (N) or PMA with U0126 (O) and immunostained for TP63 (magenta) and eGFP (green). Fin morphology is

*Figure 9 continued on next page*

*Figure 9 continued*

restored but neutrophils are still present. (**P**) Quantification of neutrophils in the fins showing U0126 does not reduce inflammation induced by loss of *hai1a* or PMA treatment. n = 8; ANOVA with Bonferroni post-test; \*\*\*p<0.001. (**Q–X**) Projected confocal images of 48hpf larval tails immunostained for β-catenin (green) and TP63 (magenta; **R, T, V, X**) of WT (**Q, R, U, V**) and *hai1a^hi2217^* (**S, T, W, X**), either untreated (**Q–T**), treated with PMA (**U, V**) or U0126 (**W, X**). (**Y–AA**) Profile plots of fluorescence distribution across cells of WT (**Y**), *hai1a^hi2217^* (**Z**), and *hai1a^hi2217^* treated with U0126 (**AA**). X-axis represents width of the cell. β-catenin immunofluorescence intensity (Y-axis) shows majority at cell edge (demarcated in light purple) in WT and rescued *hai11a* mutants, but is distributed in cytoplasm in mutant. Two cells per 3–5 larvae were analysed. Scale bars: (**B**) = 200 μm, (**F, I, K, O**) = 100 μm, (**V**) = 20 μm.

The online version of this article includes the following figure supplement(s) for figure 9:

**Figure supplement 1.** Rescue of the *hai1a* epidermal phenotype by pERK inhibitors.
**Figure supplement 2.** The barrier function of the *hai1a* epidermis is not grossly compromised.

---

using SuperScript III Reverse Transcriptase (Invitrogen) with Oligo(dT)12-18 primer. For qPCR, iTaq SYBR green (Bio-Rad) was used to amplify, with reaction dynamics measured on a Bio-Rad CFX96 Real-Time PCR Detection System. For measuring *nfkbiaa* mRNA by qPCR, the following primers (5′ to 3′) were used to amplify a region encoded on exons 4 and 5: F-AGACGCAAAGGAGCAGTGTAG, R-TGTGTGTCTGCCGAAGGTC. Reference gene was *eef1a1l1* and the primers used amplified between exon 3 to 4: F-CTGGAGGCCAGCTCAAACAT, R- ATCAAGAAGAGTAGTACCGCTAGCA TTAC.

## RNA synthesis

RNAs for *GCaMP6s* and *PKCδ-GFP* were synthesised from pCS2-based plasmids containing the respective coding sequences (*Sivak et al., 2005*; *Chen et al., 2017*). These were linearised with *Not*I (NEB), and RNA in vitro transcribed with mMESSAGE mMACHINE SP6 Transcription Kit (Ambion). RNA for *Tol2* was generated from the pT3Ts-Tol2 plasmid, linearised with *Sma*I (NEB), and transcribed with the mMESSAGE mMACHINE T3 Transcription Kit (Ambion). RNA for injection was purified by lithium chloride precipitation.

## Embryo injection and morpholino

Embryos were aligned on an agarose plate and injected at the one-cell stage with RNA or morpholino diluted in Phenol Red and Danieau's buffer using a PLI-100 microinjector (Harvard Apparatus). Injection needles were pulled from borosilicate glass capillaries (0.5 mm inner diameter, Sutter) on a Sutter P-97 micropipette puller. The Duox morpholino (AGTGAATTAGAGAAATGCACCTTTT) was purchased from GeneTools and injected at 0.4 mM with 0.2 mM of the tp53 morpholino (GCGCCA TTGCTTTGCAAGAATTG).

## TALEN mutagenesis

To generate the *ikbkg* mutant, TALEN vectors targeting the sequence ATGGAGGGCTGG in second exon were designed and constructed by ToolGen (http://toolgen.com). TALEN vectors were linearised with *Pvu*II (NEB) and purified using a PCR purification kit (Qiagen), and then used for in vitro transcription with the MEGAshortscript T7 kit (Ambion). About 170–300 pg of supplied ZFN RNAs or purified TALEN RNAs were then injected into one-cell stage WT zebrafish embryos, which were raised to 24 hr, then genomic DNA extracted.

For detection of fish with edited loci, PCR was performed on genomic DNA of injected fish with primers flanking the target site, cloned by TA cloning into pGEMT-Easy (Promega) or pCR2.1-TOPO-TA (Invitrogen) and individual clones sequenced to establish efficiency. Other embryos were raised to adulthood and their offspring were similarly genotyped to identify founder mutants.

## Small-molecule treatment

All compounds for treating embryos were dissolved in DMSO, diluted in 0.5× E2 Embryo Medium and embryos treated by immersion. The compounds, and concentrations used, with catalogue numbers were diphenyleneiodonium chloride (DPI), 40 μM (D2926, Sigma); thapsigargin, 6.25 μM (T9033, Sigma); bisindolylmaleimide I (GF109203X), 85 μM (S7208, Selleckchem); YM-254890, 32 μM

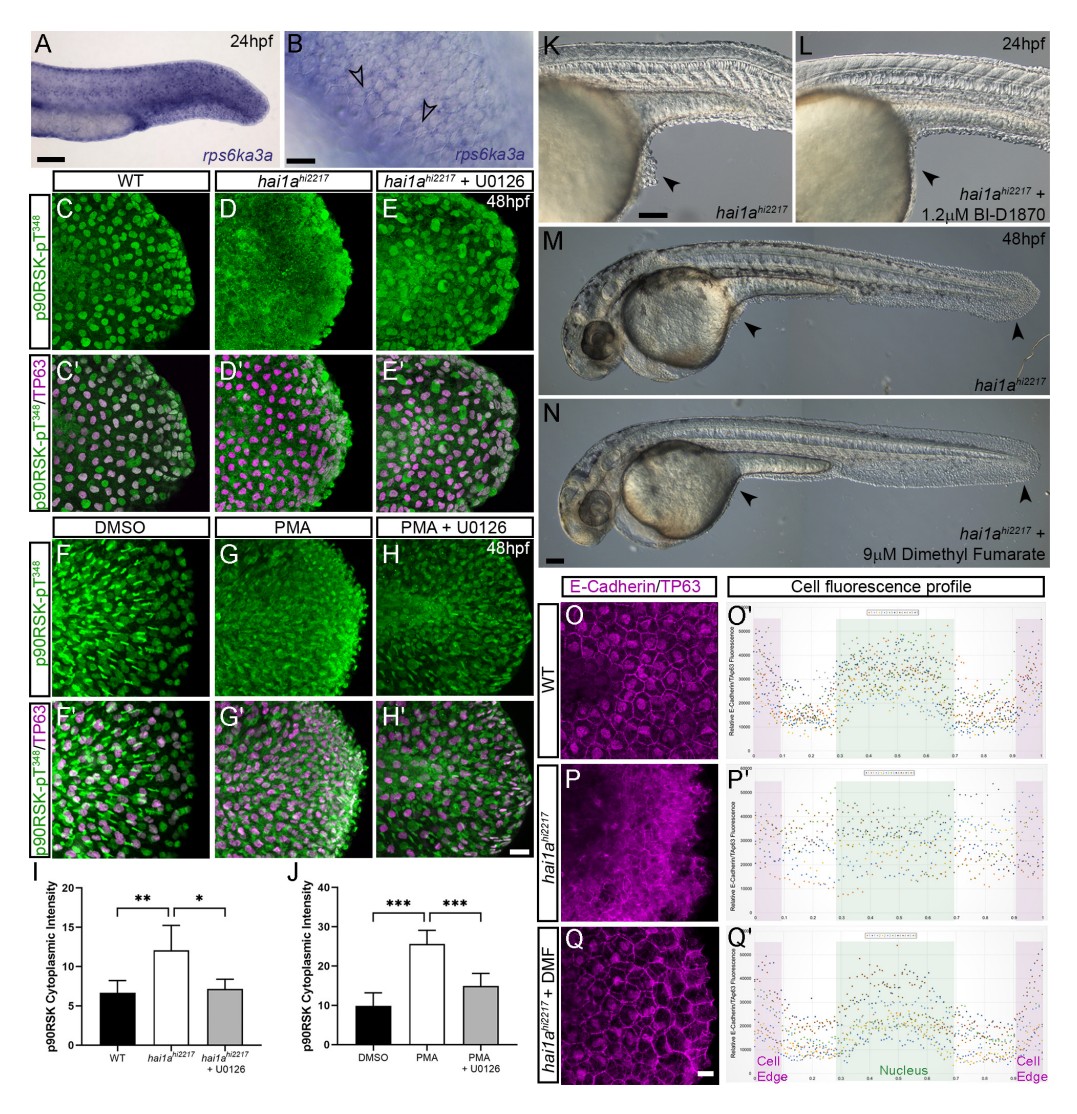

**Figure 10.** Altered RSK status in *hai1a*[hi2217] accounts for epidermal defects. (**A, B**) In situ hybridisation of *rps6ka3a* at 24hpf under low- (**A**) and high-power (**B**) magnification showing expression in basal keratinocytes. Open arrowheads in (**B**) indicate borders of EVL cells bisecting nuclei of underlying *rps6ka3a*-positive cells. (**C–H'**) Lateral projected confocal images of the tails of embryos immunostained for p90RSK (Phospho-Thr[348]) (**C–H'**) and TP63 (**C'–H'**). In both the *hai1a*[hi2217] (**D, D'**) and 125 ng/ml phorbol 12-myristate 13-acetate (PMA)-treated (**G, G'**) embryos, there is an increase in cytoplasmic levels of p90RSK (Phospho-Thr[348]) signal above the nuclear only signal seen in WT (**C, C'**) or DMSO (**F, F'**). Treatment with the pERK inhibitor U0126 reduced cytoplasmic levels but did not affect nuclear signal (**E, E'; H, H'**), (**I, J**) Quantification of immunofluorescent intensity of cytoplasmic levels of p90RSK (Phospho-Thr[348]) in basal keratinocytes of tails of 48hpf WT and *hai1a*[hi2217], treated with DMSO or U0126 (**I**), and PMA or PMA plus U0126 (**J**). Nucleus signal was excluded by masking from the DAPI channel. n = 5; t-test; ***p<0.001, **p<0.01, *p<0.05. (**K–N**) Lateral DIC images of *hai1a*[hi2217] embryos at 24hpf (**K, L**) and 48hpf (**M, N**) untreated (**K, M**) or treated with 1.2 µM BI-D1870 (**L**) or 9 µM dimethyl fumarate (DMF). Locations of epidermal aggregates and loss of tail fin morphology in *hai1a* mutants, and their rescue by RSK inhibitor treatment are indicated by arrowheads. (**O–Q**) Lateral projected confocal images of the tails of embryos immunostained with antibodies against E-cadherin and TP63 in WT (**O, P**) and *hai1a*[hi2217] treated with DMF (**Q**). (**O'–Q'**) Profile plots of fluorescence distribution across cells of WT (**O**), *hai1a*[hi2217] (**P'**), and *hai1a*[hi2217] treated with DMF (**Q'**). X-axis represents width of the cell. β-Catenin immunofluorescence intensity (Y-axis) shows majority at cell edge (E-cadherin domain demarcated in light purple) and centre of cell (nucleus demarcated in light green) in WT and rescued *hai11a* mutants, but there is no clear membrane signal in the untreated *hai1a* mutants. Two cells per five larvae were analysed. Scale bars: (**A, K, N**) = 100 µM; (**B, H**) p=20 µM.

The online version of this article includes the following figure supplement(s) for figure 10:

**Figure supplement 1.** RSK inhibitors rescue the *hai1a* phenotype.

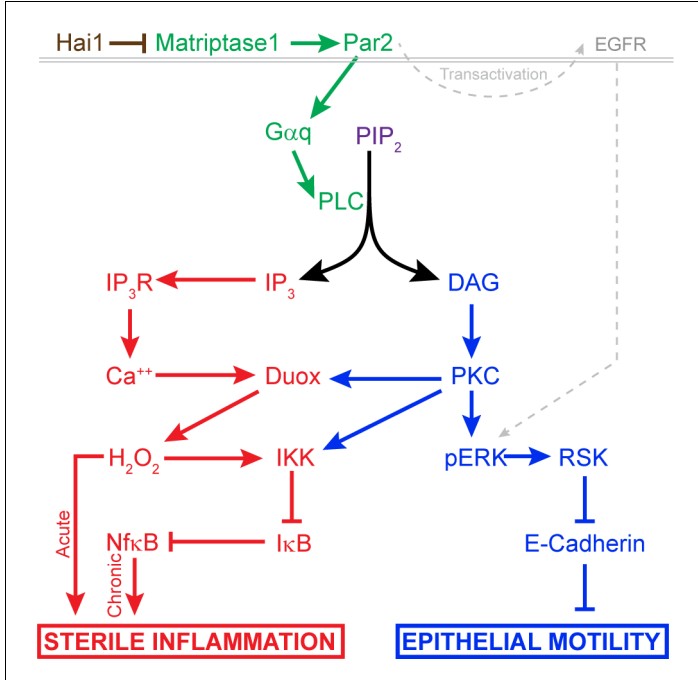

**Figure 11.** Model of pathway-activated downstream of Hai1 and Matriptase. Proposed model of pathways downstream of Hai1 which drives chronic and acute sterile inflammation (red) and epithelial motility (blue). A previously defined transactivation of EGFR is also integrated. Other pathways known to act downstream of Matriptase, involving cMet, PI3K, AKT, and mTOR, are not shown.

(10-1590-0100, Focus Biomolecules); 2-aminoethyl diphenylborinate (2-APB), 2.5 µM (D9754, Sigma), BI-D1870, 1.2 µM (Axon-1528, Axon Medchem); dimethyl fumarate, 9 µM (242926, Sigma); phorbol 12-myristate 13-acetate (PMA), 37.5 or 125 ng/ml (P8139, Sigma); U0126, 100 µM (9903, Cell Signaling Technology); PD184352 (CI-1040), 1.3 µM (S1020, Selleckchem). Unless otherwise stated, controls for all experiments were exposed to 0.5% DMSO carrier in 0.5× E2 Embryo Medium.

## Proteomic analysis

Batches of 100 WT, $ddf^{t419}$, and $ddf^{ti251}$ embryos were collected at 24 hr and 48 hr, dechorionated, deyolked, and protein extracted as per **Alli Shaik et al., 2014**. Protein was precipitated in 100% methanol at 4°C, then resuspended in 2-D cell lysis buffer (30 mM Tris-HCl, pH 8.8, containing 7 M urea, 2 M thiourea, and 4% CHAPS). 2-D DIGE and mass spectrometry protein identification was performed by Applied Biomics (Hayward, CA). Protein samples were labelled with either Cy2, Cy3, or Cy5, mixed, and then subjected to 2-D DIGE to separate individual proteins. Gels were scanned using Typhoon TRIO (Amersham BioSciences) and analysed by Image QuantTL and DeCyder (ver. 6.5) software (GE-Healthcare). Spots with more than 1.5-fold change were picked, in-gel trypsin digested, and protein identification performed by MALDI-TOF mass spectrometry and MASCOT search engine in the GPS Explorer software (Matrix Science).

## In situ hybridisation

A probe corresponding to the final 1078 bp of *rps6ka3a* (*RSK2a*; NM_212786.1) was generated by cloning a PCR-derived cDNA fragment into in pGEMT-Easy (Promega), linearising with *ApaI* (NEB) and transcribing a DIG probe with SP6 RNA polymerase (Roche). Whole-mount in situ hybridisation developed with NBT/BCIP (Roche) was performed as described (**Thisse and Thisse, 2008**).

## Immunofluorescent, dye staining, and TUNEL

For antibody staining, embryos were fixed in 4% paraformaldehyde overnight at 4°C and then washed in PBT (0.1% Triton in PBS), permeabilised in −20°C acetone for 7 min, washed in PBT,

blocked for 3 hr in Block solution (PBT supplemented with 4% BSA and 1% DMSO), then incubated overnight at 4°C with primary antibody diluted in Block solution, washed extensively in PBT, re-blocked in Block solution, then incubated overnight at 4°C with fluorescent secondary antibody diluted in Block solution. Following extensive PBT washing, embryos were cleared in 80% glycerol/PBS before imaging. Primary antibodies used and their dilutions are as follows: Chicken anti-eGFP antibody, 1:500 (ab13970, Abcam), Rabbit anti-eGFP, 1:500 (Tp401, Torrey Pines Biolabs), Rabbit anti-FITC, 1:200 (#71-1900, Thermo Fisher), Rabbit anti-beta catenin, 1:200 (ab6302, Abcam), Mouse anti-E-cadherin, 1:200 (#610181, BD Biosciences), Mouse anti-Tp63, 1:200 (CM163, Biocare Medical), Rabbit anti-phospho-p44/42 MAPK (Erk1/2) (Thr$^{202}$/Tyr$^{204}$), 1:100 (#4370, Cell Signaling Technology), Rabbit anti-p44/42 MAPK (Erk1/2), 1:100 (#9102, Cell Signaling Technology), and Rabbit anti-p90RSK (Phospho-Thr$^{348}$), 1:100 (A00487, GenScript). All secondary antibodies were purchased from Invitrogen and used at 1:700 and were Alexa Fluor-488 Donkey anti-rabbit (A21206), Alexa Fluor-647 Donkey anti-rabbit (A31573), Alexa Fluor-546 Donkey anti-mouse (A10036), and Alexa Fluor-488 Goat anti-chicken (A-11039). Nuclei were counterstained using 5 µg/ml of DAPI (4',6-diamidino-2-phenylindole, dihydrochloride; D1306, Invitrogen) added during secondary antibody incubation.

To stain hydrogen peroxide, embryos were incubated for 60 min at room temperature with 12.5 µM PFBSF (#10005983, Cayman Chemicals), then rinsed in Embryo Medium, anaesthetised, and imaged.

Fluorescent TUNEL staining was performed using the Fluorescein In Situ Cell Death Detection Kit (11684795910, Roche), with the fluorescein detected by antibody staining using rabbit anti-FITC, and co-immunostained for TP63 and eGFP. Epidermal permeability assays were conducted by immersing 36hpf embryos in 2.5 mg/ml fluorescein isothiocyanate-dextran 3–5 kDa (Sigma) or 0.075% methylene blue for 30 min and then destained in E2 medium.

## Microscopy and statistical analysis

Still and timelapse imaging was performed on upright Zeiss AxioImager M2, Zeiss Light-sheet Z.1, upright Zeiss LSM800 Confocal Microscope or Zeiss AxioZoom V16 microscopes. Embryos were mounted in 1.2% Low Melting Point Agarose (Mo Bio Laboratories) in 0.5× E2 medium in 35 mm glass-bottom imaging dishes (MatTek) or in a 1 mm inner diameter capillary for light-sheet time-lapse. When imaging was performed on live embryos, the embryo media were supplemented with buffered 0.02% Tricaine and imaging conducted at 25°C. Image processing was done using Zen 3.1 software (Zeiss), Fiji (ImageJ, ver. 1.52p), or Imaris (Bitplane) and compiled using Photoshop 2020 (Adobe). Neutrophils were tracked with TrackMate in Fiji or using the Spot function in Imaris. Kymographs were generated using the Reslice function in Fiji following generation of a line of interest across image. Fluorescence intensities were calculated using the Average Intensity function in Fiji following generation of a Region of Interest and masking of the DAPI channel to exclude the nucleus when required. In statistical analyses, n = number of embryos or cells measured, and as defined in the figure legend. GraphPad Prism was used for statistical analyses and graph generation. In all statistical tests, *p<0.05, **p<0.01, ***p<0.001. Tests used are indicated in the associated figure legend and were Student's t-test, Chi-squared test, Mann–Whitney test, or ANOVA with Bonferroni or Dunn's post-tests.

## Additional information

### Funding

| Funder | Grant reference number | Author |
|---|---|---|
| Ministry of Education - Singapore | 2015-T1-001-035 | Jiajia Ma<br>Tom J Carney |
| Ministry of Education - Singapore | MOE2016-T3-1-005 | Harsha Mahabaleshwar |

The funders had no role in study design, data collection and interpretation, or the decision to submit the work for publication.

## Author contributions
Jiajia Ma, Formal analysis, Investigation, Writing - original draft; Claire A Scott, Ying Na Ho, Ser Sue Ng, Formal analysis, Investigation; Harsha Mahabaleshwar, Data curation, Formal analysis; Katherine S Marsay, Changqing Zhang, Christopher KJ Teow, Investigation; Weibin Zhang, Resources; Vinay Tergaonkar, Sudipto Roy, Resources, Supervision; Lynda J Partridge, Enrique Amaya, Supervision; Tom J Carney, Conceptualization, Formal analysis, Supervision, Funding acquisition, Investigation, Writing - original draft, Project administration

## Author ORCIDs
Tom J Carney https://orcid.org/0000-0003-2371-1924

## Ethics
Animal experimentation: Fish were housed at the IMCB and the NTU zebrafish facilities under IACUC numbers #140924 and #A18002 respectively, and according to the guidelines of the National Advisory Committee for Laboratory Animal Research. Approval was provided by the Institutional Animal Care and Use Committees of the Biological Resource Centre (IMCB) and NTU according to Agri-Food and Veterinary Authority (AVA) Rules and the National Advisory Committee for Laboratory Animal Research (NACLAR) requirments.

## Decision letter and Author response
Decision letter https://doi.org/10.7554/eLife.66596.sa1
Author response https://doi.org/10.7554/eLife.66596.sa2

# Additional files

## Supplementary files
• Transparent reporting form

## Data availability
All data generated or analysed during this study are included in the manuscript and supporting files.

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
