## [Decision Letter]

**Acceptance summary:**

This manuscript unravels a detailed bipartite signaling mechanism, activation of which results in epithelial inflammation and cell motility. The paper is potentially of broad interest to cancer biologists and epithelial cell biologists. The data generated using the combination of genetic analyses, chemical inhibitors, and state-of-the-art confocal microscopy is of exceptionally high quality and supports the majority of the claims made in this paper.

**Decision letter after peer review:**

Thank you for submitting your article "Matriptase generates a tissue damage response via promoting Gq signalling, leading to RSK and DUOX activation" for consideration by *eLife*. Your article has been reviewed by 3 peer reviewers, including Jody Rosenblatt as the Reviewing Editor and Reviewer #1, and the evaluation has been overseen by Jonathan Cooper as the Senior Editor.

Essential Revisions:

Having conferred, we all agree that the paper is of interest and very thoroughly done regarding most aspects of the paper. The main things we would like you to address are the following:

1. Test whether the inflammatory response is a downstream consequence of poor epithelial barrier function, due to the primary epidermal phenotype. This seems to bear out in your experiments but could be addressed with dextran permeability or other approaches.

2. Address if the cell motility phenotype really lies downstream of DAG or is in a parallel pathway.

3. The paper is generally well-written but quite long. Please consider editing for conciseness and brevity.

*Reviewer #1 (Recommendations for the authors):*

1. Are the epithelial cells really more motile or is the primary cause that they no longer act collectively together like an epithelium, and migrate because of this?

2. You mention a comprehensive proteomic screen yet there is no real reference to this in how you went about it in your experiments. Were the proteomics results, in fact, used? The rationale seems to go without it. If not, do you need to include it?

3. Your analysis frequently depends on inhibitors. While you explain that sometimes morphants didn't work, I think you need to at least change the language in your results to suggest, as you could be having some off-target effects with drugs like U0126 and others.

4. Throughout, I found that the paper was logically ordered, and the results were very clear. However, I did find that the introduction and the discussion of the pathway could be more concise and clearer. It may be useful here to use a schematic to show the pathway implied before and then how your findings changed that, which could go at the beginning of the paper so people know where you are going.

5. Throughout, you mention the n in the figure legends but don't refer to what the n is-is it total embryos used for each condition or experimental numbers?

6. What are the differences between ERK activation in nucleus and cytoplasm?

*Reviewer #2 (Recommendations for the authors):*

1. It is important to directly link the hai1-matriptase system to DAG to implicate that arm of the signaling cascade. Is it possible for the authors to show the elevated levels of DAG in the hai1 mutant or to inhibit DAG in the hai1 mutant background by chemical or genetic means?

2. To support their conclusion that the two arms of the signaling cascade are causally linked with the phenotypic manifestation, the increase in H2O2 levels, Ca++ flashes, the activation of NfkB signaling, nuclear localization of phosphoRSK need to be shown just before the phenotype arise, if not for all at least for pRSK and NFkB activation. It would also be pertinent to block the signaling arms (using inhibitors) before the phenotypic onset and after the phenotype sets in to delineate the requirement of the cascade for phenotypic manifestation versus phenotypic maintenance.

3. Given that the Hai1-Matriptase system activates this bipartite signaling cascade to control the inflammation and epithelial cell motility, it is intriguing that the hai1 alleles are viable. How do authors explain the resolution of phenotype and survival of animals to adulthood? This aspect must be discussed.

*Reviewer #3 (Recommendations for the authors):*

The main problem with this manuscript lies not with the experimental data but instead with the way it is presented. The manuscript is far too long and the figures too numerous. Much of the data could be removed or moved to supplementary and the paper should be condensed down to five or six figures.

As indicated above, a major concern is that the relevance of these findings to a tissue damage response – which is openly stated in the manuscript's title – are not clear or at least not clearly supported by the data. The title of the paper is misleading as the authors don't show any real evidence that this is a true tissue damage response or show any data on how matriptase is activated (or Hai1a might be inactivated) downstream of damage/wounding. Overexpression of matriptase (or inactivation of Hai1a) leads to many of the same biological outcomes as wounding but claiming this is a tissue damage response without directly linking it to damage is a step too far and unnecessary. This message should be toned down throughout the manuscript and the title should be changed. Despite the manuscript being overly long, in multiple instances, the authors fail to adequately discuss the broader meaning of their findings, The discussion is far too long and reads more like a literature review rather than an attempt to extend their discoveries to other systems or contexts. The authors need to be much more concise throughout. If the authors can fix these problems with the structure and presentation of the paper as well as address the comments below then I would support publication.

1. To this reviewer, the authors did not clearly characterise the "inflammatory phenotype", nor explain the relevance of the "epidermal phenotype" for the fully-formed organism. As a criticism to the conclusion that neutrophils are highly migratory in Hai1a mutants, in comparison to WT, has it been considered that the increase in cell speed could be due to the lack of spatial constraint (which is instead present in WT, since neutrophils are moving within the vasculature?). Furthermore, is the lack of directionality (figure 1D) a real surprise, or a mere consequence of the lack of imposed stimuli (e.g. a wound) in these settings?

2. "We noted that again, whilst there was brief local recruitment in the mutant, more distant neutrophils appeared apathetic to the wound and remained migrating randomly, unlike in wild-type where neutrophils were recruited from a great distance and tracked to the wound with high directionality". The distance traveled by neutrophils cannot be appreciated from figures 1F-G.

3. Using the stain pentafluorobenzenesulfonyl fluorescein (PFBSF) (Maeda et al., 2004), we observed significantly higher levels of H2O2 in the trunk and tails of hai1a mutants at 24 and 48hpf (Figure 1G-I, Figure S1G-H). These figure references are misplaced.

4. In figure 4, the authors demonstrate that Hai1a mutants have hyperactivation of the NFkB signalling (Figure 4B), and that this is due to H2O2, as it can be rescued by treatment with a DUOX morpholino (Figure 4K). Strangely however, they also found that inhibition of calcium flashes via 2-APB only very moderately ameliorates the NFkB overactivation (Figure S4F-G). If DUOX is activated by Calcium, shouldn't the two treatments (DUOX morpholino and 2-APB treatment) yield the same response? How do the authors explain this observation, and fit it within their working model?

5. The authors briefly mention that in a subset of cells (TP63 +ve cells), pERK was observed both in the nucleus and cytosol, while in another subset of cells (peridermal cells) it was only observed in the nucleus. As in other instances throughout the manuscript, the authors fail to further comment on this observation. What is the implication for such different subcellular localisation? Are, for instance, post-translation modifications or trafficking throughout the cells different in the two subsets of cells? (Figure 8)

6. In multiple parts of the manuscript the authors hint at the likelihood of a cross-talk between the two identified signaling pathways, as these may well be co-responsible for the inflammatory phenotype. This is for instance the case when they show that treating embryos with a PKC inhibitor ameliorates the epidermal phenotype but only slightly the inflammatory phenotype. This is an interesting observation, but to this reviewer the authors failed to adequately discuss it both throughout the Results section and in the discussion.

---

## [Author Response]

Essential Revisions:Having conferred, we all agree that the paper is of interest and very thoroughly done regarding most aspects of the paper. The main things we would like you to address are the following:1. Test whether the inflammatory response is a downstream consequence of poor epithelial barrier function, due to the primary epidermal phenotype. This seems to bear out in your experiments but could be addressed with dextran permeability or other approaches.

We have tested this using fluorescein dextran and methylene blue permeability assays (Zhang, J et al., (2015) Exp Dermatol, 24: 605-610; Richardson, R., et al. (2013) The Journal of Investigative Dermatology, 133(6), 1655–1665). Whilst larval fin wounds show strong uptake of the dyes, we were unable to show robust staining in *hai1a* mutants. This data has been added as Figure 9 – —figure supplement 2. The lack of overt dye penetration made it difficult to draw conclusions from this as it is still possible there is permeability problems, just that we cannot detect it through these assays. We think more pertinent was the fact that we can rescue the epithelial defects robustly (eg with MAPK or PKC inhibition) without completely rescuing the inflammation phenotype, which you would expect if inflammation was purely a consequence of epithelial defects. Finally, new data added in response to a request by reviewer #2 has shown that the increase in Ca^++^ and H2O2 occurs very early, prior to epidermal phenotypic presentation (New panels 1L and 3H). As these are well described pro-inflammatory drivers in zebrafish, we believe this adds to the case that the inflammation is, to some extent, independent of the epithelial defects. We thus think it unlikely that inflammation in *hai1a* is solely due to epithelial defects. HOWEVER, there is clear evidence from us and others that PKC and MAPK activation can also promote inflammation. We have added remarks on this to the discussion.

2. Address if the cell motility phenotype really lies downstream of DAG or is in a parallel pathway.

We have used a GFP-tagged PKCδ construct (Sivak, et al. (2005). Dev Cell, 8(5), 689-701) to show that PKC is relocated to plasma and nuclear membranes in *hai1a* from a homogeneous cytoplasmic distribution in WT (new panels 7B and Figure 7 – —figure supplement 1). This relocation assay is commonly used for demonstrating increased DAG in cells, as the activity of PLC cleaves PIP2 and increases DAG levels in the membranes. In addition, we have previously show that Gq inhibitors rescue the epithelial phenotype, which would be expected to block PLC activity and thus reduce IP3 and DAG levels, whilst PMA phenocopies the epithelial defects. Finally, inhibition of the DAG receptor, PKC, also strongly rescues the epithelial defects. Thus, these four independent experiments make a strong case for DAG contributing to the epithelial phenotype.

3. The paper is generally well-written but quite long. Please consider editing for conciseness and brevity.

Agreed. We apologise for the initial length of the manuscript. We have rationalised and streamlined the introduction and discussion to improve clarity and conciseness. We have shortened the Introduction from 781 words to 540, whilst the Discussion has been revised from 1578 to 1307 words yet has also incorporated the more detailed discussions on the topics required by the reviewers. We think the new flow has made it much clearer and improved readability. We have removed the emphasis on tissue damage in the discussion as suggested by Reviewer 3. We are also now under the *eLife* 5000-word recommendation for Research Articles.

Reviewer #1 (Recommendations for the authors):1. Are the epithelial cells really more motile or is the primary cause that they no longer act collectively together like an epithelium, and migrate because of this?

It’s difficult for us to say at this point. The extent of active cytoskeletal rearrangement in *hai1a* mutants has never been directly assessed as far as we are aware, but RSK is known to activate GEFs such as LARG (see discussion). Further it is known that *hai1a* mutants have increased MMPs (LeBert, et al. (2015). Development 142(12), 2136–2146). Our data and videos certainly show loss of adherens junctions and cells migrate over each other which has prompted us to label this behaviour simply as mesenchymal. We point out that this is unlikely to be due to a full EMT process as our previous analysis failed to show LOSS of E-Cadherin levels, rather just relocation. This is consistent with the know role of RSK in p120 phosphorylation.

2. You mention a comprehensive proteomic screen yet there is no real reference to this in how you went about it in your experiments. Were the proteomics results, in fact, used? The rationale seems to go without it. If not, do you need to include it?

This 2D gel proteomic screen identified Peroxiredoxin-4 as having altered pI, and this was our critical inroad into the mechanism by highlighting likely H2O2 changes. We have now included the full reports for this screen in supplementary data for the readers.

3. Your analysis frequently depends on inhibitors. While you explain that sometimes morphants didn't work, I think you need to at least change the language in your results to suggest, as you could be having some off-target effects with drugs like U0126 and others.

We have now added qualifiers to our conclusion statements throughout.

4. Throughout, I found that the paper was logically ordered, and the results were very clear. However, I did find that the introduction and the discussion of the pathway could be more concise and clearer. It may be useful here to use a schematic to show the pathway implied before and then how your findings changed that, which could go at the beginning of the paper so people know where you are going.

Yes – agreed. It was quite impenetrable. Sorry! We have re-written the introduction and discussion to streamline and include only the relevant background information. I think this has made the main contributions of the paper to the field much clearer and more accessible. I believe now that the simplified introduction has obviated the need for a schematic in the introduction, but happy to draft one if you think it is still required above the summary diagram Figure 11. The verbosity of the paper was a common theme from the reviewers, and we are happy to take further reviewer and editor guidance on this if our edited version is still deemed to be too long. We are now under the 5000 suggested length for *eLife* Research Articles.

5. Throughout, you mention the n in the figure legends but don't refer to what the n is-is it total embryos used for each condition or experimental numbers?

Generally, n was number of embryos for each condition, except where we have counted a number of cells in a range of embryos, where we have been explicit about this in the figure legend. We have now added this explanation to the Materials and methods.

6. What are the differences between ERK activation in nucleus and cytoplasm?

Activated pERK1/2 are known to have phosphorylation targets at the membrane (eg EGFR), in the cytoplasm (eg p90RSK) and in the nucleus (eg transcription factor Ets). The mechanisms regulating sub-cellular localisation, and thus availability of pERK for phosphorylating substrates, is a large field of its own, but levels of scaffolding proteins, activating kinases and phosphorylation of nuclear pore proteins likely contribute (Pouysségur J, Volmat V, Lenormand P. Biochem Pharmacol. 2002 Sep;64(5-6):755-63.). We suspect that there will also be other ERK mediated changes in different cell types in *hai1a*, including modulation of other cytoplasmic targets and transcriptional effects. We have added a sentence to the discussion highlighting this.

Reviewer #2 (Recommendations for the authors):1. It is important to directly link the hai1-matriptase system to DAG to implicate that arm of the signaling cascade. Is it possible for the authors to show the elevated levels of DAG in the hai1 mutant or to inhibit DAG in the hai1 mutant background by chemical or genetic means?

As described above in the response to Essential Revision #2, we have now included data on the GFP-tagged PKCδ construct (Sivak, et al. (2005). Dev Cell, 8(5), 689-701) to show that PKC is relocated to plasma and nuclear membranes in *hai1a* from a homogeneous cytoplasmic distribution in WT (new panels 7B and Figure 7 – —figure supplement 1). This is a common assay for increased DAG levels in cells.

We now have 4 main results supporting our case that DAG contributes to the cell motility phenotype of *hai1a* mutants.

a. PKCδ-GFP relocates to nuclear and plasma membranes from the cytoplasm in *hai1a* as expected for increase in cellular DAG levels.

b. Gq, activator of PLC, rescues the epithelial defects, whilst the IP3R inhibitor does not. By elimination this suggests the other product of PLC activity, DAG, as responsible for the epithelial defects

c. Inhibition of the DAG receptor, PKC, rescues the epithelial defects

d. A DAG mimic, PMA, phenocopies the hai1a epithelial phenotypes.

2. To support their conclusion that the two arms of the signaling cascade are causally linked with the phenotypic manifestation, the increase in H2O2 levels, Ca++ flashes, the activation of NfkB signaling, nuclear localization of phosphoRSK need to be shown just before the phenotype arise, if not for all at least for pRSK and NFkB activation. It would also be pertinent to block the signaling arms (using inhibitors) before the phenotypic onset and after the phenotype sets in to delineate the requirement of the cascade for phenotypic manifestation versus phenotypic maintenance.

We have now included early staining and videos at 15-17hpf to look at the levels of these signals prior to phenotype presentation. These analyses required us to genotype embryos after staining and imaging as the epithelial defects were not apparent at this early stage. We have found that prior to phenotype manifestation:

a. H2O2 is highly increased in the yolk area at 16hpf (Figure 1K-L)

b. Calcium flashes are seen at 16hpf extensively over the yolk epidermis (Video 4; Figure 3G-H)

c. Nfkb is NOT increased noticeably at 15hpf, suggesting that this might be a driver of later chronic inflammation unlike the known early role of H2O2. (Figure 4 – —figure supplement 1C-D)

d. pERK levels are increased at early stages in nascent aggregates (Figure 8 – —figure supplement 1H) and in regions prior to forming aggregates (Figure 8G)

e. Similarly, cytoplasmic levels of pRSK are increased at 17hpf in regions outside of aggregate formation (Figure 10 – —figure supplement 1A-C)

We have compared U0126 treatment from 16hpf with later treatment from 26hpf and see that the latter gives a milder rescue. We interpret this to mean that the epithelial phenotype in *hai1a* mutants requires sustained pERK activity (Figure 9 – —figure supplement 1K-M’).

3. Given that the Hai1-Matriptase system activates this bipartite signaling cascade to control the inflammation and epithelial cell motility, it is intriguing that the hai1 alleles are viable. How do authors explain the resolution of phenotype and survival of animals to adulthood? This aspect must be discussed.

Apologies for the confusion here, the strong alleles (*fr26, ti257, t419)* are early lethal and die within the first week. The mild allele *hi2217* is semi-viable to a highly variable extent. We used to see adult survival, but less so know, and it is common for them to die at 1 month. We think that there are modifiers that alter severity. The reason some mild *hai1a^hi2217^* mutants survive has been recently determined to be due to cell extrusion and entosis (see Armistead et al. 2020). We have clarified this in the introduction.

Reviewer #3 (Recommendations for the authors):The main problem with this manuscript lies not with the experimental data but instead with the way it is presented. The manuscript is far too long and the figures too numerous. Much of the data could be removed or moved to supplementary and the paper should be condensed down to five or six figures.As indicated above, a major concern is that the relevance of these findings to a tissue damage response – which is openly stated in the manuscript's title – are not clear or at least not clearly supported by the data. The title of the paper is misleading as the authors don't show any real evidence that this is a true tissue damage response or show any data on how matriptase is activated (or Hai1a might be inactivated) downstream of damage/wounding. Overexpression of matriptase (or inactivation of Hai1a) leads to many of the same biological outcomes as wounding but claiming this is a tissue damage response without directly linking it to damage is a step too far and unnecessary. This message should be toned down throughout the manuscript and the title should be changed.

This is quite a reasonable suggestion, and I did get a little carried away with this topic. So I have changed the title and significantly reduced the emphasis on tissue damage. I point out however that we are not the first to make this claim about what the phenotypes of zebrafish *hai1a* represent, with the Coughlin lab concluding that these appear as tissue injury responses (Schepis, A., et al. (2018). J Cell Biol, 217(3), 1097-1112), whilst our original analysis of the *hai1a* mutant also discussed the phenotype in the context of wounding (Carney, T. J., et al. (2007). Development, 134(19), 3461-3471). I have referenced the Schepis paper specifically where I have briefly alluded to this topic. However, I fully agree that the discussion inappropriately overemphasised this. I hope this reads far more balanced now.

Despite the manuscript being overly long, in multiple instances, the authors fail to adequately discuss the broader meaning of their findings, The discussion is far too long and reads more like a literature review rather than an attempt to extend their discoveries to other systems or contexts. The authors need to be much more concise throughout. If the authors can fix these problems with the structure and presentation of the paper as well as address the comments below then I would support publication.1. To this reviewer, the authors did not clearly characterise the "inflammatory phenotype", nor explain the relevance of the "epidermal phenotype" for the fully-formed organism. As a criticism to the conclusion that neutrophils are highly migratory in Hai1a mutants, in comparison to WT, has it been considered that the increase in cell speed could be due to the lack of spatial constraint (which is instead present in WT, since neutrophils are moving within the vasculature?). Furthermore, is the lack of directionality (figure 1D) a real surprise, or a mere consequence of the lack of imposed stimuli (e.g. a wound) in these settings?

We think it unlikely that the epidermis constrains neutrophils or retards their migration to any appreciable extent. Evidence for this can be seen in the WT fin wounding in Video 2, where at 178mins, a neutrophil appears from the anterior and tracks under the epidermis towards the wound with great speed. It is difficult to envisage that the epidermis this far from the wound is compromised and permitting increased neutrophil speed. Further to this reviewer’s concerns, we looked at 2 mutants with subepidermal blistering of the fin folds immediately under the posterior cardinal vein (*pif* and *nel*). This blistering alone did not induce extravasation of neutrophils out of the vasculature, except when the fin degenerated to cause a localised wound, towards which they migrated very specifically (Author response image 1).

**Author response image 1. sa2fig1:** Epidermal detachment does not lead to increased neutrophil motility. A-E: Lateral confocal images of the tails of 36hpf WT (A), *hai1a^hl227^* (B), *pif^tm95b^* (C), and *nel^tq207^* (D-E) immunofluorescently stained for Mpx (Green) and superimposed on the DIC channel. The sub-lamina densa blistering of the fins in *pif* and *nel* mutants does not lead to the epidermal inflammation seen in *hai1a* mutants. Only where there is epidermal damage and degeneration are neutrophils seen outside the vasculature, and then they only migrate to the damage site (E, red arrowhead). Scale bar A = 100μm.

2. "We noted that again, whilst there was brief local recruitment in the mutant, more distant neutrophils appeared apathetic to the wound and remained migrating randomly, unlike in wild-type where neutrophils were recruited from a great distance and tracked to the wound with high directionality". The distance traveled by neutrophils cannot be appreciated from figures 1F-G.

We have now included graphical representation of this data as a new panel (Figure 10 – —figure supplement 1D).

3. Using the stain pentafluorobenzenesulfonyl fluorescein (PFBSF) (Maeda et al., 2004), we observed significantly higher levels of H2O2 in the trunk and tails of hai1a mutants at 24 and 48hpf (Figure 1G-I, Figure S1G-H). These figure references are misplaced.

Apologies. These figure references have now been corrected.

4. In figure 4, the authors demonstrate that Hai1a mutants have hyperactivation of the NFkB signalling (Figure 4B), and that this is due to H2O2, as it can be rescued by treatment with a DUOX morpholino (Figure 4K). Strangely however, they also found that inhibition of calcium flashes via 2-APB only very moderately ameliorates the NFkB overactivation (Figure S4F-G). If DUOX is activated by Calcium, shouldn't the two treatments (DUOX morpholino and 2-APB treatment) yield the same response? How do the authors explain this observation, and fit it within their working model?

Our model (Figure 11) also includes activation of Duox by PKC which has been described previously (Rigutto et al. JBC 284(11), 6725-6734). This reconciles the ability of PMA to generate H2O2, (Figure 6M) and NfkB (Figure 6U) but not calcium (Figure 6Q, S). It also explains why IP3R inhibition has a strong effect on H2O2 but only a moderate effect on NfKb levels (Figure 3J, Figure 4 – —figure supplement 1I). We have included this in the discussion.

5. The authors briefly mention that in a subset of cells (TP63 +ve cells), pERK was observed both in the nucleus and cytosol, while in another subset of cells (peridermal cells) it was only observed in the nucleus. As in other instances throughout the manuscript, the authors fail to further comment on this observation. What is the implication for such different subcellular localisation? Are, for instance, post-translation modifications or trafficking throughout the cells different in the two subsets of cells? (Figure 8)

Reviewer #1 also had this query and I refer to the answer above. In truth, we have no data on this at all. We note that there are differences in reliance on ErbB2 function between the cells and this may translate to different pERK behaviours, but this is just a guess. Briefly there are transcriptional, membrane and cytoplasmic targets of pERK phosphorylation. Subcellular localisation of pERK, and thus accessibility to substrates, is determined by amount of scaffolding proteins, activating kinases and phosphorylation of nuclear pore proteins. The full picture of how pERK is localised within the cell is quite poorly understood (Pouysségur J, Volmat V, Lenormand P. Biochem Pharmacol. 2002 Sep;64(5-6):755-63.). Recently calcium levels are described to alter pERK subcellular localisation (Chuderland D, Marmor G, Shainskaya A, Seger R. Cell Physiol Biochem. 2020 May 12;54(3):474-492). We have now added comments on this to the discussion, but I have to confess that at his point, we are unclear on what this means in terms of the biology, beyond the RSK target.

6. In multiple parts of the manuscript the authors hint at the likelihood of a cross-talk between the two identified signaling pathways, as these may well be co-responsible for the inflammatory phenotype. This is for instance the case when they show that treating embryos with a PKC inhibitor ameliorates the epidermal phenotype but only slightly the inflammatory phenotype. This is an interesting observation, but to this reviewer the authors failed to adequately discuss it both throughout the Results section and in the discussion.

We have now discussed this in the discussion, and integrated this into our model clearly, justifying it through referring to our own observations and also by referencing the literature. We have now also included reference to (Feng et al. (2010) PLoS Biol, 8(12), e1000562), which showed that RasV12 transformed zebrafish keratinocytes also show increased hydrogen peroxide release and inflammation.